

# Experimental interpretation of adequate weight-metric combination for dynamic user-based collaborative filtering

Savas Okyay[1,2,*] and Sercan Aygun[3,4,*]

[1] Computer Engineering, Eskisehir Osmangazi University, Eskisehir, Turkey
[2] Computer Engineering, Eskisehir Technical University, Eskisehir, Turkey
[3] Computer Engineering, Yildiz Technical University, Istanbul, Esenler, Turkey
[4] Electronics Engineering, Istanbul Technical University, Istanbul, Maslak, Turkey
* These authors contributed equally to this work.

Corresponding author
Sercan Aygun, ayguns@itu.edu.tr

## ABSTRACT

Recommender systems include a broad scope of applications and are associated with subjective preferences, indicating variations in recommendations. As a field of data science and machine learning, recommender systems require both statistical perspectives and sufficient performance monitoring. In this paper, we propose diversified similarity measurements by observing recommendation performance using generic metrics. Considering *user-based* collaborative filtering, the probability of an item being preferred by any user is measured. Having examined the best neighbor counts, we verified the test item bias phenomenon for similarity equations. Because of the statistical parameters used for computing in a global scope, there is implicit information in the literature, whether those parameters comprise the focal point user data statically. Regarding each dynamic prediction, user-wise parameters are expected to be generated at runtime by excluding the item of interest. This yields reliable results and is more compatible with real-time systems. Furthermore, we underline the effect of significance weighting by examining the similarities between a user of interest and its neighbors. Overall, this study uniquely combines significance weighting and test-item bias mitigation by inspecting the fine-tuned neighborhood. Consequently, the results reveal adequate *similarity weight* and *performance metric* combinations. The source code of our architecture is available at https://codeocean.com/capsule/1427708/tree/v1.

## INTRODUCTION

Recommender systems (RS) are utilized in various applications, and users interact with them on a range of application-specific platforms. Personal data and previous activities are combined to understand a user's taste. Any recommendation of items on a platform can be provided. Considering both online and offline applications, a stable architecture is essential for machine learning and for promoting the business of any platform. Recommender systems have been implemented on various platforms, including social media (*Kazienko, Musiał & Kajdanowicz, 2011*), healthcare (*Calero Valdez & Ziefle, 2019*), journals (*Wang et al., 2018*), music (*Andjelkovic, Parra & O'Donovan, 2019*; *Celma &*

*Herrera, 2008*), suggestion systems, and movie recommendation frameworks (*Moreno et al., 2013*; *Isinkaye, Folajimi & Ojokoh, 2015*; *Wang, Wang & Xu, 2018*). Recommender systems focus on analyzing preferences and deciding the prospective action of a user. A person performs activities (such as passing remarks, leaving comments, giving rates, liking, or disliking products) on a specific application because these activities are all logged into a database. Movie-based RS has been the focus of many data scientists for two significant reasons. First, scientific datasets such as *MovieLens* (*Grouplens, 1992*) and *Netflix* (*Netflix, 2009*) are readily available and easy to use. Second, an overall RS architecture has been established that is entirely compatible with additional user and item features, enabling scientists to measure two well-known phenomena in collaborative filtering (CF): (i) *user-based* similarity and (ii) *item-based* similarity.

This study aims to measure the effect of correlation adjustment. Considering the literature, there are many reported RS implementations; however, it is unclear how the inclusion or exclusion of a test item is determined during statistical parameter computations. Similarity calculations between two users require analytical computations, such as the *mean* and *median*. Theoretical studies may set statistical arguments as global parameters to encapsulate computations for all upcoming test attempts concerning time complexity and memory management. Although setting parameters globally is computation-friendly, if all statistical primitives are set in this wide scope, test items become less dependent on the related test attempts. Hence, the expected recommendation may be slightly false. Considering real-time applications, the rating value of the recommended item is unknown. In this study, the dynamic effect of the *item-of-interest* (*IOI*) is examined to demonstrate the difference between theoretical and real-time performance. In addition, the *co-rated item count* (*CIC*) between the user-of-interest and its neighbors is utilized to revise the calculated similarity weight for further constant multiplications (*Ghazanfar & Prugel-Bennett, 2010*; *Levinas, 2014*; *Zhang & Yuan, 2017*; *Gao et al., 2012*; *Bellogín, Castells & Cantador, 2014*; *Zhang et al., 2020*). Thus, the correlation between users is connected to the commonly rated item counts. We refer to this multiplication as the *CIC-based significance weighting (SW)* method by demonstrating its performance. We interpret the efficiency of the *IOI* and *SW* conditions based on four different similarity equations, including *Pearson similarity*, *median-based robust correlation*, *cosine similarity*, and *Jaccard similarity*.

In general, studies have focused on finding ways to increase the efficiency of RS. The closer the forecast is to the user preference obtained, the more accurate the system design is. However, the performance metrics of a system can be more than a single prediction accuracy. In this study, we examined previously proposed similarity equations and performance metrics. The research constructs a perspective on how to connect user similarity measurements to an enlarged number of performance metrics, including those from other disciplines. *Schröder, Thiele & Lehner (2011)* propose the utilization of relatively less known metrics such as *informedness*, *markedness*, and *Matthews correlation* because they are superior to *precision*, *recall*, and *F1-measure*. *Schröder, Thiele & Lehner (2011)* acknowledge that these performance metrics are suitable for determining the

top-*n* recommendation in e-commerce applications; therefore, we evaluate the performance of these metrics compared to the well-known ones.

Previous RS implementations have either a relatively small set of metrics for testing (*Feng et al., 2018*; *Bag, Kumar & Tiwari, 2019*; *Li et al., 2014*; *Nguyen et al., 2020*) or a limited range of specific parameters, such as the *best neighborhood* (*Ghazanfar & Prugel-Bennett, 2010*; *Arsan, Koksal & Bozkus, 2016*; *Sánchez et al., 2008*; *Liu et al., 2013*; *Huang & Dai, 2015*; *Sun et al., 2017*). Any user is provided with a recommendation by examining the closest neighbors who have the same tendencies for the related *IOI*. Instead of setting the best neighbor count (*BNC*) to a constant value, the neighborhood should be appropriately determined. Therefore, we parameterize the number of neighbors, s.t., using $\varepsilon$ step size between the least neighbor count (*LNC*) and most neighbor count (*MNC*).

A comprehensive back-end software architecture has been developed in this study. Our framework[1] is an adaptive tool that enables the test environment to capture the general behavior of high-density datasets with an adjusted $\varepsilon$.

To the best of our knowledge, no previous studies on RS have extensively focused on an adequate combination of similarity measurements and performance metrics. Overall, the following highlights are presented in the scope of this study.

- We construct an RS framework that highlights the possible pitfalls and enhancements in RS architectural designs. Therefore, the following two perspectives are applied to the similarity equations.

  ○ The first perspective underlines the *dynamicity* principles of real-time systems by excluding the *IOI*, known as **no item-of-interest** (*nIOI*).
  ○ The second perspective emphasizes the results of the utilization of significant weights. Considering the *SW* method, the more common the rating counts from neighbors observed are, the more significant the weights.

- The *BNC* is analyzed and determined experimentally considering a number of performance metrics.
- Extensive tests are applied to popular *MovieLens* releases with randomized trials of separate runs.
- Considering the evaluation, relatively less known performance metrics such as *informedness*, *markedness*, *and Matthews correlation* are examined comprehensively. In addition, established metrics such as *precision*, *sensitivity*, *specificity*, *F1-measure*, *fallout*, *miss rate*, etc., including error metrics, are compared. These prediction-oriented metrics have been extensively demonstrated with notable outcomes.
- *Prevalence threshold* and *threat score*, which are frequently practiced in other disciplines, are analyzed in the context of RS.

- Finally, the heat-map tables for the top-performing *BNC*s connected to the adequate weight-metric combinations are presented.

[1] The open-source code information is available in the Data Availability statement. Any dataset can be analyzed as long as it meets the requirement of *user × item* matrix format.

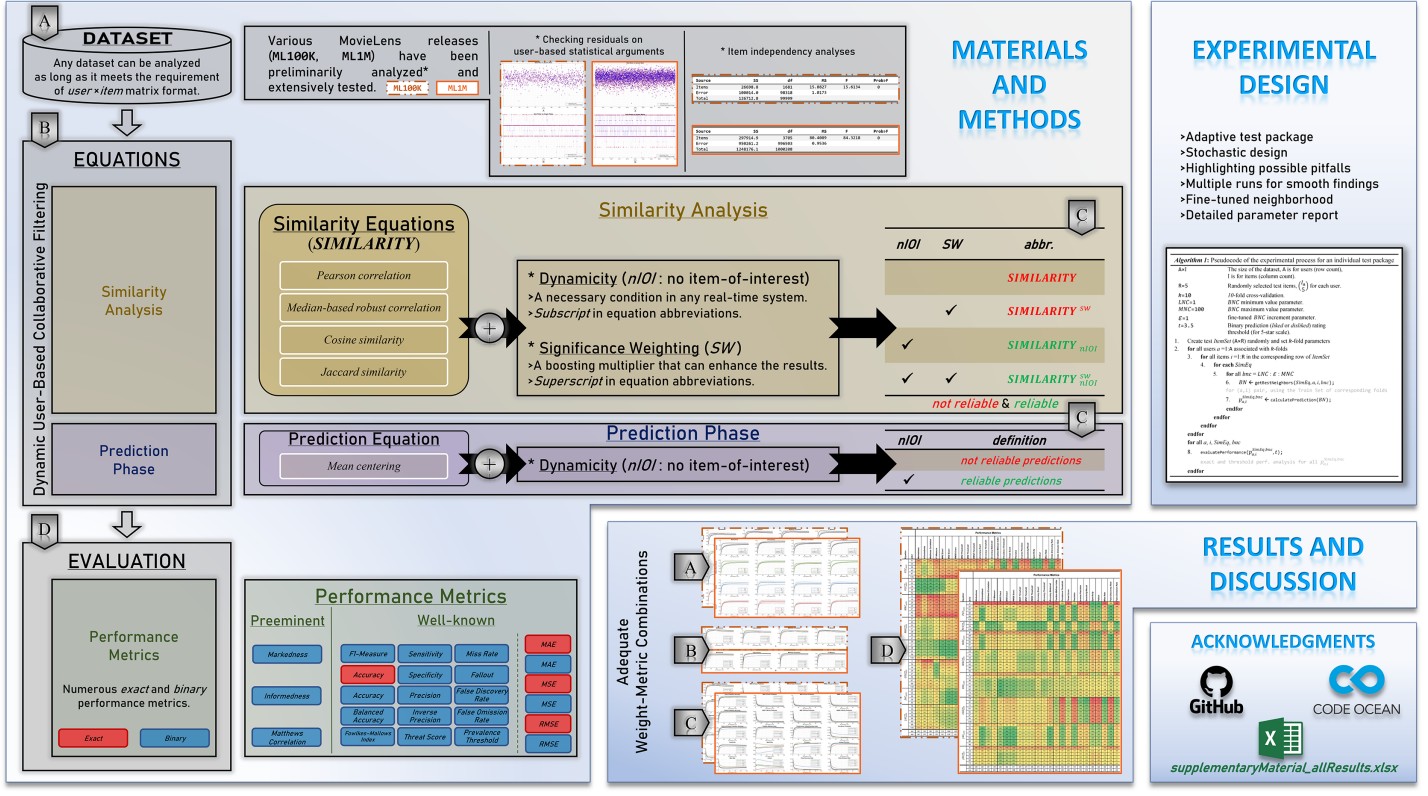

**Figure 1** Depiction of the paper structure, technical summary, and contributions.

The brief of the overall paper structure and each subsection content are visually presented in Fig. 1. The remainder of this study is organized as follows. The materials and methods are provided in Section "Materials and Methods", wherein the nomenclature and dataset details are presented. In addition, similarity equations and performance metrics used throughout this study are presented in the same section. Section "Experimental Design" includes the details of the computation environment and preliminary selection of top-performing neighbors. This is followed by the extensive results in Section "Results and Discussion". In the last section, the conclusion and recommendations for future research are presented.

## MATERIALS AND METHODS

This section describes the dataset used and the methods applied. First, the technical details of *MovieLens* releases are provided. Thereafter, the touchstone similarity equations and the modifications, considering the *nIOI* phenomenon and *SW*, are discussed. Finally, the performance metrics implemented in the proposed RS framework are presented. The symbols and abbreviations used throughout this study are listed in Table 1.

### A. The MovieLens

Considering the current RS applications, the basic practical structure of data commonly has a *user × item* matrix format. One of the frequently trained scientific datasets is

**Table 1 Symbols and abbreviations list.**

| Symbol/Abbreviation | Explanation |
| --- | --- |
| $a$ | User-of-interest |
| $\hat{a}$ | User-of-interest where test item bias is discarded |
| $i$ | Any item-of-interest |
| $U$ | Possible nominees to be a neighbor |
| $u$ | Any possible neighbor for collaboration |
| $u^*$ | Sorted and selected neighbor |
| $\mathbf{r_u}$ | Rating vector of $u$ for all items |
| $r_{u,i}$ | Rating of $u$ for $i$ |
| $\bar{r}_u$ | The mean of given ratings for $u$ |
| $\tilde{r}_u$ | The median of given ratings for $u$ |
| $I_u$ | The rating history of $u$ |
| PCC | Pearson Correlation Coefficient |
| MRC | Median-based Robust Correlation coefficient |
| COS | COSine similarity |
| JAC | JACcard similarity |
| $w_{a,u}^S$ | Similarity weight between $a$ and $u$ for equation $S$, where $S$ can be given similarity equations |
| $p_{a,i}$ | The rating prediction of $a$ for $i$ |
| CIC | Co-rated Item Count between two users |
| TP | True Positive |
| TN | True Negative |
| FP | False Positive |
| FN | False Negative |

*MovieLens* (*Harper & Konstan, 2015*) which has several releases based on size and additional content.

Considering Table 2, the main types of *MovieLens* can be reviewed depending on the rating size. For example, ML100K has 100,000 clicks. The *MovieLens* dataset is upgraded several times for the expanded types and for the versions of previous releases. For instance, the ML100K type has various releases, such as one that includes one to five ratings with only decimal values. The latest ML100K version consists of 0.5 steps between ratings, including *half stars*. However, this version is not recommended for shared research results because it is a developing dataset. Several previous studies focused on the tried-and-trusted original ML100K release, which is a pioneering collection and has considerably efficient runtime performance. Considering the scope of this study, we utilize this original release, which includes only full stars. Additionally, we encapsulated extensive experiments of ML1M to maintain full-star rating scaling parallelism with the ML100K. Therefore, we comparatively present the results related to the original ML100K and ML1M.

In this section, preliminary dataset analyses are presented. To validate the methods used in the following sections, residual checks on user-based statistical arguments and item-based independence analyses were performed as follows.

Table 2 MovieLens (ML) release comparison (*Grouplens, 1992*; *Harper & Konstan, 2015*).

| Releases | ML100K | ML1M | ML10M | ML20M | ML25M |
|---|---|---|---|---|---|
| Number of Ratings | 100,000 | 1,000,209 | 10,000,054 | 20,000,263 | 25,000,095 |
| Number of Users | 943 | 6,040 | 69,868 | 138,493 | 162,541 |
| Number of Movies | 1,682 | 3,706 | 10,681 | 27,278 | 62,423 |
| Timespan | 09/1997–04/1998 | 04/2000–02/2003 | 01/1995–01/2009 | 01/1995–03/2015 | 01/1995–11/2019 |
| Miscellaneous Information | At least 20 ratings by each user, simple demographic information for users (age, gender, occupation, zip-code). 5-star rating. | At least 20 ratings by each user, simple demographic information for users (age, gender, occupation, zip-code). 5-star rating. | At least 20 ratings by each user. No demographic information. Each user is represented by only an ID. 5-star rating with half-stars. | At least 20 ratings by each user. No demographic information. Each user is represented by only an ID. 5-star rating with half-stars. | At least 20 ratings by each user. No demographic information. Each user is represented by only an ID. 5-star rating with half-stars. |

(1) Checking residuals on user-based statistical arguments

The dynamicity effect of user-based statistical arguments (such as the mean and median) is discussed in this subsection. As static and dynamic approaches are the main focus, a visualization of their residual analysis on the utilization of the arguments is presented. Therefore, the rating history of each user was examined based on the statistical observations. Considering any user, it is observed that the statistical values of all the ratings change when an item is assumed as unrated. The *IOI* values that are individually excluded from the user vector are dynamically processed. The effect of each discarded rating was recorded as a residual over the dynamic mean or median. Thereafter, the static observation and dynamic approach were evaluated using the residual approach.

Figures 2 and 3 show the static and dynamic analyses based on the (A) mean and (B) median usage based on the ML100K and ML1M releases, respectively. The *x*- and *y*-axis show the user ID and unique rating values, respectively. Each red dot statically indicates the mean or median values of all user ratings, whereas the blue dots show the deviation of the unit ratings from the static value. It may be observed in the median analysis that the blue dots were aggregated, while the outliers in the datasets were suppressed, indicating the superiority of the median over the mean in the presence of outliers.

(2) Item independency analyses

Considering the dynamic approach regarding real-time systems, excluding the *IOI* in the users' statistical calculations depends on the item's independence condition. Therefore, each particular item in the datasets was analyzed based on the independence. The leave-item-out approach emerges as a useful method because the items are independent of each other. Consequently, an item-based one-way analysis of variance (ANOVA) was performed. Each column (*i.e.*, each item) was subjected to testing in the *user × item* matrix,

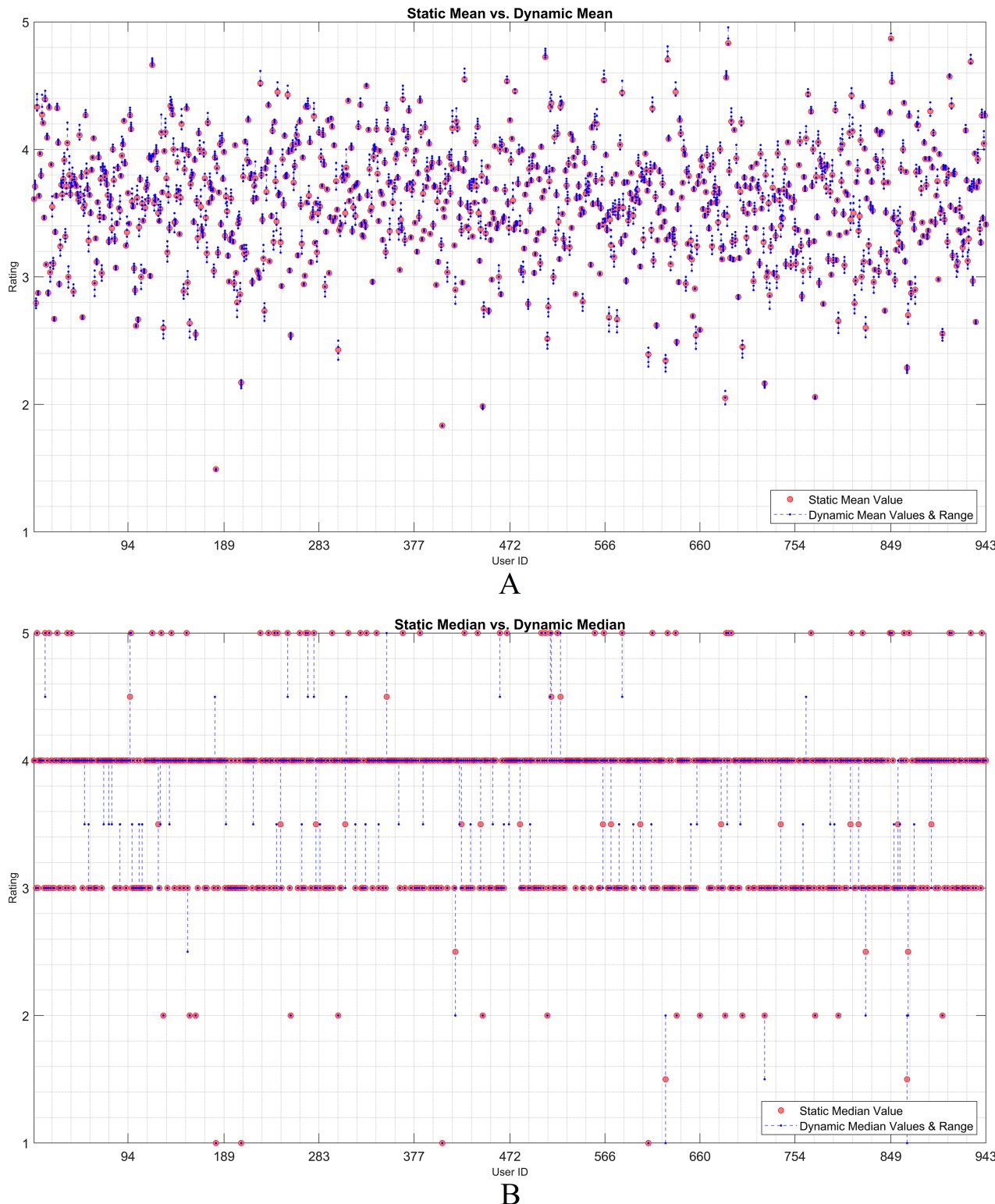

**Figure 2** User-based (A) mean & (B) median residuals on static *vs.* dynamic conditions: ML100K.

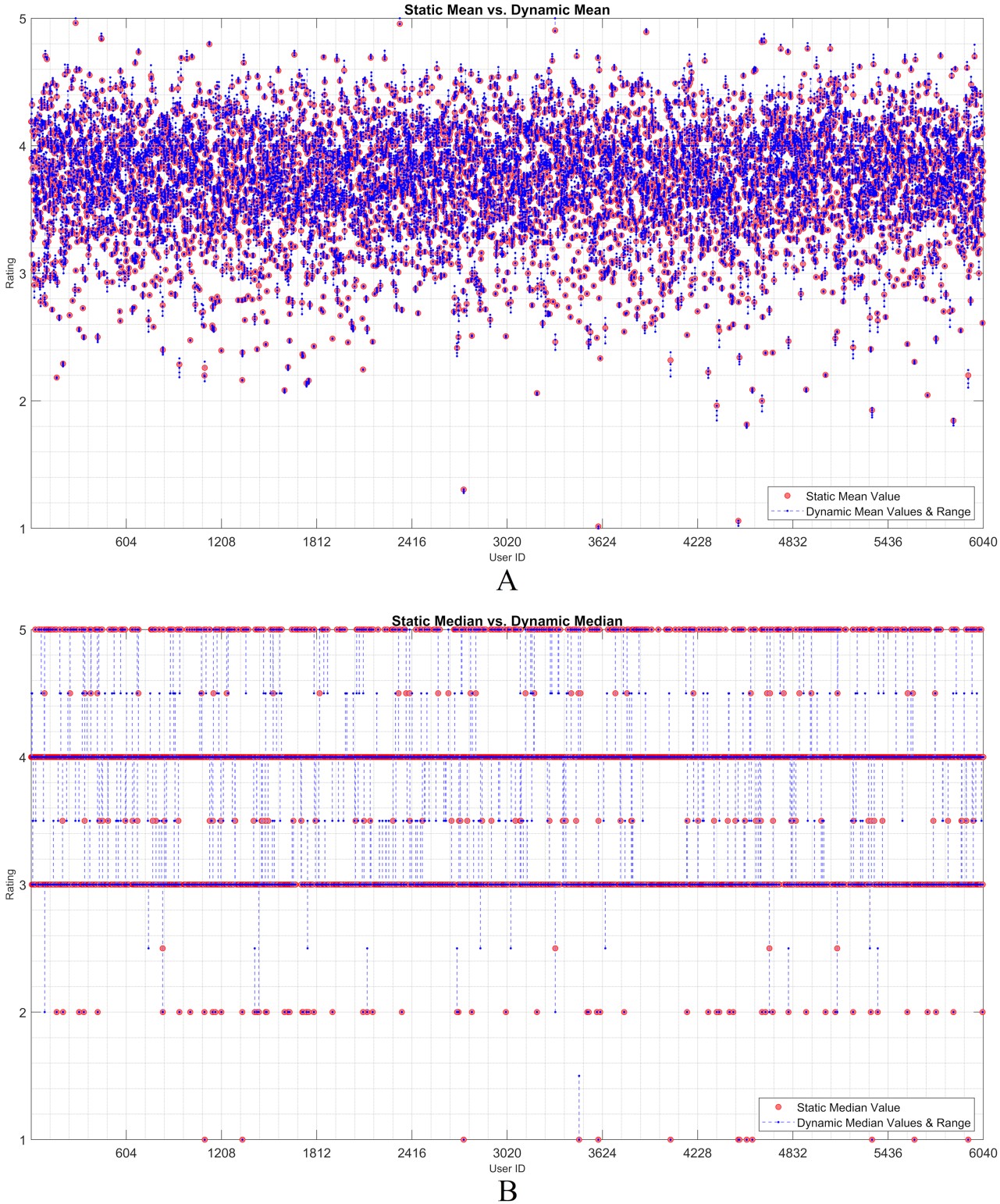

**Figure 3  User-based (A) mean & (B) median residuals on static *vs.* dynamic conditions: ML1M.**

**Table 3 ANOVA table for ML100K.**

| Source | SS | df | MS | F | Prob>F |
|--------|-----|-----|-----|-----|--------|
| Items | 26,698.8 | 1,681 | 15.8827 | 15.6134 | 0 |
| Error | 100,014.0 | 98,318 | 1.0173 | | |
| Total | 126,712.8 | 99,999 | | | |

**Table 4 ANOVA table for ML1M.**

| Source | SS | df | MS | F | Prob>F |
|--------|-----|-----|-----|-----|--------|
| Items | 297,914.9 | 3,705 | 80.4089 | 84.3218 | 0 |
| Error | 950,261.2 | 996,503 | 0.9536 | | |
| Total | 1,248,176.1 | 1,000,208 | | | |

validating their independencies. The ANOVA provides information about inter- and intra-group variations. By calculating the sum of squares (SS), degrees of freedom (df), and mean squared errors (MS), the F-test (the ratio of inter- and intra-group variability) is applied. Considering Tables 3 and 4, an analysis of the ML100K and ML1M releases is presented. The validity of item independence is proven by both the $F \gg 1$ and probability (P) values, which are obtained from the F-distribution. The lower the P-values, the higher the chances of strong evidence against the *null hypothesis*. P-values $\ll 0.05$ (significance level) were obtained, indicating that the *null hypothesis* is rejected.

## B. Similarity and prediction equations

The four touchstone similarity equations and prediction formula are considered in this section. Before the technical statements, an overview of the application perspective of the touchstone equations is provided.

*PCC* is within the scope of several studies. Music RS is among the most common applications (*Kuzelewska & Ducki, 2013*). In particular, considering music genre recommendations, *PCC* has attracted attention. Additionally, there are other applications. Mukaka explains the management of medical data based on the utilization of *PCC* (*Mukaka, 2012*), as in other studies (*Akoglu, 2018*; *Miot, 2018*). Apart from these, book recommendations *via PCC* (*Kurmashov, Latuta & Nussipbekov, 2016*; *Sivaramakrishnan et al., 2018*), e-commerce applications (*Lee, Park & Park, 2008*), and academic paper RS (*Lee, Lee & Kim, 2013*) are intriguing alternatives. Other *PCC* examples can be found in *Adiyansjah, Gunawan & Suhartono (2019)*, *Cataltepe & Altinel (2009)*, *Sigg (2009)* and *Shepherd & Sigg (2015)*. The most common application is movie-based RS (*Dhawan, Singh & Maggu, 2015*; *Madadipouya, 2015*; *Sheugh & Alizadeh, 2015*), which is the motivation of the present work. Movie genre correlations were calculated using *PCC* by *Kim et al. (2010)*. *Hwang et al. (2016)* also presented the details of the *PCC*, considering movie genre classification. Nonetheless, *PCC* has some disadvantages, considering the linear averaging procedures. *Tan & He (2017)* indicated the underlying limitations of the *PCC* in reinforcing the effect of correlation. Subsequently, they proposed the resonance

similarity between users by parametrizing the median of the rating values. They constructed a physical analogy between user similarities regarding simple harmonic motion in a coordinate system (*Tan & He, 2017*). The mean of the rating vectors can be vulnerable to outliers and biases, as *Garcin et al. (2009)* emphasized the superiority of median-based rating aggregations over mean- and mode-based aggregations. Therefore, the use of *MRC* was proposed to suppress outliers in user ratings in the context of RS. *COS* is another frequently used method in RS owing to its simple calculation, and is encountered in movie-related applications (*Singh et al., 2020*; *Wahyudi, Affandi & Hariadi, 2017*), research paper recommendation (*Philip, Shola & John, 2014*; *Ahmad & Afzal, 2020*; *Samad et al., 2019*), cognitive similarity-based design (*Nguyen et al., 2020*), article suggestion system (*Rajendra, Wang & Raj, 2014*), and music RS (*Aiolli, 2013*). Furthermore, *JAC* has been evaluated in several studies (*Bag, Kumar & Tiwari, 2019*; *Sun et al., 2017*; *Meilian et al., 2014*; *Rana & Deeba, 2019*). *JAC* has an essential feature regarding binary rating analysis (*Zahrotun, 2016*) and is considered a measure that does not treat absolute ratings (*AL-Bakri & Hashim, 2019*). From the perspective of merits and demerits of the relevant touchtone equations, the following detailed information is introduced. In general, *PCC* provides a concept of the presence, absence, and degree of correlation. It also provides feedback on positive and negative correlations. However, computationally *PCC* is complex owing to the complex algebraic requirements in its formula. It is also ineffective against outlier values. One merit of *MRC* emerges with its median usage, whereas *PCC* is based on the assumption of only linear correlation and is not the appropriate option for homogeneous data. Although *MRC* shows similar features to *PCC*, it is a more suitable option, especially for data with outliers. However, demerits of *PCC* are also valid for *MRC*. Using angle information, *COS* easily calculates the correlation between data, which are even quite far in terms of Euclidean distance. However, *COS* does not provide a concept about magnitude. *Saranya, Sudha Sadasivam & Chandralekha (2016)* emphasized that *COS* is ineffective in capturing similar users who rated quite few items. Conversely, *JAC* has the merits of binary set processing. The utilization of *JAC* is simple because the equation requires only the set operations, especially for the binary ratings. However, if a system has vectors of categorical or multiple-valued data, then *JAC* requires a preprocessing step for the binarization (*Supriya*; *Saranya, Sudha Sadasivam & Chandralekha, 2016*).

Together with the proposed *nIOI* and *SW* modifications over similarity equations, different combinations are interpreted by inferring the underlying affinity. Subsequently, the *PCC*, *MRC*, *COS*, and *JAC* similarities are stated technically. Owing to the various performance metrics presented in Subsection "Performance metrics", adequate weight-metric combinations are to be determined.

(1) Pearson correlation

Pearson correlation is an acclaimed measure adopted in many data-mining approaches that address the similarity of measurement data. Regarding the user-based CF, the *PCC* is a tool used to define in-between user similarity by considering the item ratings. Pearson

weighs all connected neighbors and calculates the degree of a linear relationship between two users. Thus, a weight for each correlated neighbor is derived, achieving a linear relationship by processing the deviation from the mean values, $\bar{r}$ (*Pearson, 1894*). Considering Eq. (1), the similarity formula between two users, *a* and *u*, is indicated.

$$w_{a,u}^{PCC} = \frac{\sum_{i \in (I_a \cap I_u)} \left( (r_{a,i} - \bar{r}_a) \times (r_{u,i} - \bar{r}_u) \right)}{\sqrt{\sum_{i \in (I_a \cap I_u)} (r_{a,i} - \bar{r}_a)^2} \times \sqrt{\sum_{i \in (I_a \cap I_u)} (r_{u,i} - \bar{r}_u)^2}}. \tag{1}$$

(2) Median-based robust correlation

Median-based robust correlation is a method that replaces the linear *mean* procedures with the *median* operation (*Shevlyakov, 1997*; *Shevlyakov & Smirnov, 2011*; *Pasman & Shevlyakov, 1987*). The utilization of the averages may suffer from the *skewness* problem (*Pearson, 1895*; *Sato, 1997*). In addition, outliers can affect mean values. The *MRC*, which has the median of rating values instead of the averages similar to those in *PCC*, represents the suppression of outliers in the ratings of each user. Considering Eq. (2), the *MRC* formula is as follows.

$$w_{a,u}^{MRC} = \frac{\sum_{i \in (I_a \cap I_u)} \left( (r_{a,i} - \tilde{r}_a) \times (r_{u,i} - \tilde{r}_u) \right)}{\sqrt{\sum_{i \in (I_a \cap I_u)} (r_{a,i} - \tilde{r}_a)^2} \times \sqrt{\sum_{i \in (I_a \cap I_u)} (r_{u,i} - \tilde{r}_u)^2}}. \tag{2}$$

Contrary to the mean values of user ratings, $\bar{r}_a$ and $\bar{r}_u$ in Eq. (1), the median values, $\tilde{r}_a$ and $\tilde{r}_u$, represent the midpoints of the sorted ratings. The formula is similar to *PCC*, and the median point of the user ratings is considered as a neutral mark.

(3) Cosine similarity

Another similarity is based on the *cosine* function. By performing the Euclidean dot product, the cosine value between the two *n*-element vectors $A$ and $B$ can be determined. Thus, the similarity is based on $A.B = \|A\|\|B\| cos\Theta$. Considering the *user-based* similarity calculation *via COS*, the similarity weight between *a* and *u* can be measured as in Eq. (3).

$$w_{a,u}^{COS} = \frac{r_a \cdot r_u}{\|r_a\| \, \|r_u\|}. \tag{3}$$

Some other versions of the conventional *COS* have also been developed, such as the *adjusted* (*Gao, Wu & Jiang, 2011*) and *asymmetric cosine similarities* (*Aiolli, 2013*).

(4) Jaccard similarity

Jaccard similarity is a measure of common elements in two sets. The rating history of a user under test, $I_a$, and corresponding neighbor history, $I_u$, are calculated using Eq. (4). The *JAC* considers the two sets by having the ratio of their *intersection* to the *union*. The range of this similarity coefficient is $0 \leq w_{a,u} \leq 1$, where zero indicates that there are

no common elements, whereas one implies that all the elements in the two sets are fully joint.

$$w_{a,u}^{JAC} = \frac{|I_a \cap I_u|}{|I_a \cup I_u|}.$$  (4)

(5) Prediction equation

After the similarity calculations for all the best neighbor nominees, the obtained weights are sorted by denoting $w_{a,u^*}$. Thereafter, considering the sorted weights and *BNC* limits, the best neighbors are determined. The prediction phase must be completed to achieve the recommendation score. The rating prediction formula, which is known as the *mean centering* approach (*Saric, Hadzikadic & Wilson, 2009*; *Zeybek & Kaleli, 2018*; *Sarwar et al., 2001*; *Wu et al., 2013*; *Singh et al., 2020*), is given in Eq. (5).

$$p_{a,i} = \bar{r}_a + \frac{\sum_{u^*=1}^{BNC} \left( \left( r_{u^*,i} - \bar{r}_{u^*} \right) \times w_{a,u^*} \right)}{\sum_{u^*=1}^{BNC} w_{a,u^*}}.$$  (5)

## C. Modified equations

Considering the equations given in the previous section, we modified these formulas. As a result, the efficiency of RS was significantly improved under some circumstances.

The modifications were made based on two aims, including (i) to create a system model suitable for real-time applications and (ii) to boost the similarity weights. The former is related to the dynamicity, whereas the latter is related to the user-of-interest and its neighbor by considering their *CIC* as a constant multiplier, thereby signified weights can be obtained.

Considering the first phenomenon, the already rated test item was discarded from the user-of-interest rating history to predict the actual rating. Thus, during the *mean* or *median* calculations in formulas such as *PCC* and *MRC*, the item is excluded as expected in real-time systems. This case is also valid for other measurements, such as *COS* and *JAC*, where the related item is removed from the vectors in progress. To indicate this phenomenon, we use the *nIOI* subscript by denoting $\hat{a}$ in the equations. As explained in the first section, the negligence of the *nIOI* in many other RS applications is thought to be due to runtime concerns in vast scientific tests.

The second phenomenon is relative weight scaling, known as *SW*. This gives priority to a neighbor with more common ratings for the items. After calculating the co-rated item count, the weights in similarity calculations are signified using $CIC = |I_a \cap I_u|$, a constant multiplier (*Okyay & Aygün, 2020*). There are other alternatives in the literature (*Ghazanfar & Prugel-Bennett, 2010*; *Levinas, 2014*; *Zhang & Yuan, 2017*; *Gao et al., 2012*; *Bellogín, Castells & Cantador, 2014*; *Zhang et al., 2020*; *Okyay & Aygün, 2020*). For instance, *Bellogín, Castells & Cantador (2014)* compared different *user-user* weighting schemes. That set in this study is known as *user overlap*, which calculates common item counts between user neighbors (*Raeesi & Shajari, 2012*), considering *Herlocker* (*Herlocker,*

*Konstan & Riedl, 2002*) and *McLaughlin's significance weightings* (*McLaughlin & Herlocker, 2004*) together with *trustworthiness* (*Weng, Miao & Goh, 2006*) and *trust deviation* (*Hwang & Chen, 2007*). However, they either include extra parameters or require complex computations. *Raeesi & Shajari (2012)* compared the *SW* strategies by underlining the *user overlap*, which demonstrated a higher efficiency, considering the error rates, although there were few arguments to process.

Regarding the modifications, each equation from the previous section is updated using the two abovementioned phenomena. Considering *PCC*, based on Eq. (1), Eq. (6) is obtained by excluding test-item bias. Subsequently, Eq. (7) is the signified version of Eq. (1) by applying only *SW*.

$$w_{a,u}^{PCC_{nIOI}} = \frac{\sum_{i \in (I_{\hat{a}} \cap I_u)} \left( (r_{\hat{a},i} - \bar{r}_{\hat{a}}) \times (r_{u,i} - \bar{r}_u) \right)}{\sqrt{\sum_{i \in (I_{\hat{a}} \cap I_u)} (r_{\hat{a},i} - \bar{r}_{\hat{a}})^2} \times \sqrt{\sum_{i \in (I_{\hat{a}} \cap I_u)} (r_{u,i} - \bar{r}_u)^2}}, \tag{6}$$

$$w_{a,u}^{PCC^{sw}} = |I_a \cap I_u| \times w_{a,u}^{PCC}. \tag{7}$$

The same approach is followed for the *MRC* in the Eq. (8). The *SW* multiplication is expressed in Eq. (9).

$$w_{a,u}^{MRC_{nIOI}} = \frac{\sum_{i \in (I_{\hat{a}} \cap I_u)} \left( (r_{\hat{a},i} - \tilde{r}_{\hat{a}}) \times (r_{u,i} - \tilde{r}_u) \right)}{\sqrt{\sum_{i \in (I_{\hat{a}} \cap I_u)} (r_{\hat{a},i} - \tilde{r}_{\hat{a}})^2} \times \sqrt{\sum_{i \in (I_{\hat{a}} \cap I_u)} (r_{u,i} - \tilde{r}_u)^2}}, \tag{8}$$

$$w_{a,u}^{MRC^{sw}} = |I_a \cap I_u| \times w_{a,u}^{MRC}. \tag{9}$$

Regarding the *COS*, Eq. (10) shows the vector operations of the ratings in which the test-item bias is discarded, and the *SW* approach is described in Eq. (11).

$$w_{a,u}^{COS_{nIOI}} = \frac{\boldsymbol{r_{\hat{a}}} \cdot \boldsymbol{r_u}}{\|\boldsymbol{r_{\hat{a}}}\| \, \|\boldsymbol{r_u}\|}, \tag{10}$$

$$w_{a,u}^{COS^{sw}} = |I_a \cap I_u| \times w_{a,u}^{COS}. \tag{11}$$

Finally, *JAC* with the modifications is shown in Eq. (12) and (13).

$$w_{a,u}^{JAC_{nIOI}} = \frac{|I_{\hat{a}} \cap I_u|}{|I_{\hat{a}} \cup I_u|}, \tag{12}$$

$$w_{a,u}^{JAC^{sw}} = |I_a \cap I_u| \times w_{a,u}^{JAC}. \tag{13}$$

Although previous studies (*Ghazanfar & Prugel-Bennett, 2010*; *Levinas, 2014*; *Zhang & Yuan, 2017*; *Gao et al., 2012*; *Bellogín, Castells & Cantador, 2014*; *Raeesi & Shajari, 2012*; *Herlocker, Konstan & Riedl, 2002*; *McLaughlin & Herlocker, 2004*; *Weng, Miao & Goh, 2006*; *Hwang & Chen, 2007*) have presented a good understanding of *SW* using different perspectives, there is a lack of detailed performance analyses in the relevant literature. We contribute to the relative comparison of similarity equations enhanced with *SW*, including their corresponding performance.

The two phenomena *nIOI* and *SW* were independently measured. Subsequently, to monitor the hybrid effect of these approaches, both are utilized by obeying the generalized formula in Eq. (14).

$$w_{a,u}^{SIMILARITY_{nIOI}^{SW}} = |I_{\hat{a}} \cap I_u| \ \times \ w_{a,u}^{SIMILARITY_{nIOI}}. \tag{14}$$

The modified rating prediction formula is given by Eq. (15). Considering the *nIOI*, $\bar{r}_a$ is updated compared with the original equation in Eq. (5).

$$p_{a,i} = \bar{r}_{\hat{a}} + \frac{\sum_{u^*=1}^{BNC} \left( (r_{u^*,i} - \bar{r}_{u^*}) \times w_{a,u^*} \right)}{\sum_{u^*=1}^{BNC} w_{a,u^*}}. \tag{15}$$

## D. Performance metrics

The final phase of the proposed RS design involves monitoring the running algorithm. Because the CF is an intersection of statistics and machine learning, conclusive information on its performance is necessary. Particularly, understanding the inter-relational achievement of similarity equations with implied modifications requires thorough performance monitoring through numerous metrics. Regarding this, we focus on two main groups: *well-known metrics* and *preeminent metrics*. The former includes the frequently practiced performance monitoring, whereas the latter is less known in the literature, but still prominent for RS.

(1) Well-known metrics

The well-known metrics in Table 5 are applied to the framework to provide insight for further studies. The explanations of the listed metrics are briefly summarized as follows.

First, *exact accuracy* is a metric used to measure the exact matches of actual ratings ($r_{a,i}$) and corresponding predictions ($p_{a,i}$). The accuracy computation is considered for the predicted rating, controlling whether $p_{a,i} = r_{a,i}$ or $p_{a,i} \neq r_{a,i}$. Considering the frameworks that use $N$-scale ratings, *exact accuracy* can provide a precise observation. In addition, *threshold accuracy* has also been used following the binary decision of *liked* and *disliked* items. By denoting $r_{a,i} \in \mathbb{N}^+$ or $r_{a,i} \in \mathbb{R}^+$ where $argmax(r_{a,i}) = N$ over $N$-scale ratings, $t \in \mathbb{R}^+$ as a threshold value should be set satisfying $t < N$. Thereafter, the rating value $r$ compared to the threshold value $t$ is evaluated to label *liked* or *disliked* items in a binary sense (*Bag, Kumar & Tiwari, 2019*).

Second, correctly predicted positive values are measured *via sensitivity*, which is also known as the *recall* or *true positive rate*. In addition, the measure of the actual positives is monitored by *precision*, namely the *positive predictive value*. *Precision* is referred to by *Powers (2007)* as the *true positive accuracy*, indicating the confidence score. From *sensitivity* and *precision*, the *F1-measure* is calculated using the harmonic mean. On the

**Table 5  Well-known performance metrics.**

| Metric name | Formula |
| --- | --- |
| *Exact Accuracy* | $\frac{\textit{Exact Prediction Count}}{TP + TN + FP + FN}$ |
| *Threshold Accuracy* | $\frac{TP + TN}{TP + TN + FP + FN}$ |
| *Sensitivity/Recall/True Positive Rate* | $\frac{TP}{TP + FN}$ |
| *Precision/Positive Predictive Value* | $\frac{TP}{TP + FP}$ |
| *F1-Measure* | $\frac{2 \times TP}{2 \times TP + FP + FN}$ |
| *Specificity/Inverse Sensitivity/True Negative Rate* | $\frac{TN}{FP + TN}$ |
| *Inverse Precision/Negative Predictive Value* | $\frac{TN}{TN + FN}$ |
| *False Discovery Rate* | $1 - Precision$ |
| *False Omission Rate* | $1 - Inverse\ Precision$ |
| *Fallout/False Positive Rate* | $1 - Specificity$ |
| *Miss Rate/False Negative Rate* | $1 - Sensitivity$ |
| *Fowlkes–Mallows Index* | $\sqrt{Precision \times Sensitivity}$ |
| *Balanced Accuracy* | $\frac{(Sensitivity + Specificity)}{2}$ |
| *Threat Score/Critical Success Index* | $\frac{TP}{TP + FN + FP}$ |
| *Prevalence Threshold* | $\frac{\sqrt{Sensitivity \times (1 - Specificity)} + Specificity - 1}{Sensitivity + Specificity - 1}$ |

contrary, aside from the positive decisions made, negatives have also been considered. Thus, *specificity* (or *inverse sensitivity*) represents the proportion of real negative cases. Moreover, *inverse precision* (or *negative predictive value*), which is also known as the *true negative value* by *Powers (2007)*, shows the predicted negative instances. The *false discovery* and *omission rates* were deduced complementarily from the maximum metric scores of the *precision* and *inverse precision*. *Shani & Gunawardana (2011)* assert that the *false discovery rate* can be an alternative control mechanism, which is the proportion of *FP* to the actual positives. Similarly, the *false omission rate* is the ratio of *FN* to all negatives (*Mukhtar et al., 2018*).

Sensitivity and *specificity* are attributed as the *true positive* and *negative rates*, respectively. Similarly, the *fallout* and *miss rate* represent the *false positive* and *false negative rates*, respectively. The irrelevant recommendation ratio is obtained *via* the *fallout*. The *miss rate* is the ratio of the items that are not recommended although they are relevant. A recent study performed *fallout* and *miss rate* by practicing a personalized nutrition recommendation study (*Devi, Bhavithra & Saradha, 2020a*). Similar to the *F1-measure*, the utilization of *precision* and *recall* by means of geometric mean also appears in the *Fowlkes–Mallows index*. Considering another recent study, *Panda, Bhoi & Singh (2020)* discussed how to increase the *Fowlkes–Mallows index* similar to the *F1-measure*. *Balanced accuracy* provides a better perspective for performance analyses, considering an imbalanced confusion matrix. *Balanced accuracy* is the arithmetic mean of *sensitivity* and *specificity*. To understand the algorithm efficiency, utilizing several metrics, such as *balanced accuracy*, results in considerable feedback.

**Table 6 Preeminent performance metrics.**

| Metric name | Formula |
| --- | --- |
| *Markedness* | $\frac{TP}{TP+FP} + \frac{TN}{TN+FN} - 1$ |
| *Informedness* | $\frac{TP}{TP+FN} + \frac{TN}{TN+FP} - 1$ |
| *Matthews Correlation* | $\frac{TP \times TN - FP \times FN}{\sqrt{(TP+FP) \times (TP+FN) \times (TN+FP) \times (TN+FN)}}$ |

The final metrics are the *threat score* and *prevalence threshold*. The former considers *hits*, *misses*, and *false alarms* in the confusion matrix (*Hogan et al., 2010*). The latter emphasizes a sharp change in the *positive predictive value*. The *prevalence threshold* with a more geometric interpretation of the performance measurement with a focus on *positive* and *negative predictive values* is provided in *Balayla (2020)*, and it has been applied to test analyses of Covid-19 screening (*Balayla et al., 2020*).

The metrics constructed from the confusion matrix and the error metrics, such as *mean absolute error* (*MAE*), *mean squared error* (*MSE*), and *root mean squared error* (*RMSE*), are considered in this study. *Li et al. (2014)*, in their privacy-preserving CF approach, measured performance using *RMSE* and *MAE* as in *Nguyen et al. (2020)*. The *RMSE* has demonstrated its efficiency in measuring error performance. For instance, considering the *Netflix Prize* competition, it was used as a vital indicator of the implementation (*Bell & Koren, 2007*).

(2) Preeminent metrics

Although previous studies have given priority to *F1-measure*, *precision*, *recall*, and error-based measures (*Bag, Kumar & Tiwari, 2019*; *Li et al., 2014*; *Nguyen et al., 2020*; *Hong-Xia, 2019*), some other performance metrics which are relatively less recognizable in RS also provide robust decisions. According to *Chaaya et al. (2017)*, well-known metrics cause significant biases, and *markedness*, *informedness*, and *Matthews correlation* are noteworthy alternatives. Because these preeminent metrics have limited utilization in the literature, they have been considered with a priority in this study. They consist of confusion matrix primitives as shown in Table 6. The definitions of these metrics are briefly reviewed below.

(i) Markedness

The proportion of correct predictions is measured by *markedness*. This metric is free from an unbalanced confusion matrix. The *markedness* scores in the range [−1, +1] and its associated formula is demonstrated in Eq. (16). *Markedness* can be a substitution of *precision*, which can be used as a tool that shows the status of the recommendation and "chance" (*Schröder, Thiele & Lehner, 2011*). To the best of our knowledge, *markedness* is one of the least considered metrics in the literature in the scope of RS science, although it supremely supplies information related to *positive* and *negative predictive values*. For instance, this phenomenon is known as *DeltaP* in the field of psychology, and Powers

confirms that *markedness* is considered as a good predictor of human associative judgments (*Powers, 2007*; *Shanks, 1995*).

$$Markedness = Precision + Inverse\ Precision - 1. \tag{16}$$

(ii) Informedness

The second preeminent metric is *informedness*, which includes *sensitivity* and its *inverse* as shown in Eq. (17). *Informedness* scores in the same range $[-1, +1]$, as in *markedness* (*Schröder, Thiele & Lehner, 2011*). This metric is also known as the *Youden's index* because it differs from the *accuracy*, considering imbalanced events in the confusion matrix. The returned score defines a perfect prediction by +1, or indicates the opposite by −1 (*Broadley et al., 2018*). The efforts of *informedness* in RS science are limited although there are intriguing applications that practice this promising metric. *Pilloni et al. (2017)* performed *informedness* in e-health recommendations. Considering hotel recommendations, *informedness* has also been used to check the performance of the multi-criteria system. *Ebadi & Krzyzak (2016)* set two performance metrics as *prediction-* and *decision-based*, where *informedness* was considered in the scope of decision-based metrics. Regarding another research field, *Marciano, Williamson & Adelman (2018)* utilized *informedness* in the context of genetic applications. This was considered as the relative level of confidence (*Marciano, Williamson & Adelman, 2018*). In addition, *Layher, Brosch & Neumann (2017)* measured the performance of neuromorphic applications by assessing *informedness*.

$$Informedness = Sensitivity + Inverse\ Sensitivity - 1. \tag{17}$$

(iii) Matthews correlation

The *Matthews correlation* is a promising observation of binary labeling. Considering Eq. (18), a wide implicit observation is obtained with a score in the range of $[-1, +1]$. The interpretation of this metric considers the three focus points: *perfect prediction*, *random prediction*, and *total disagreement* between the actual and predicted values. According to the score range, each corresponding focus point is indicated by +1, 0, and −1, respectively (*Boughorbel, Jarray & El-Anbari, 2017*).

$$Matthews\ Correlation = \sqrt{\begin{array}{c} Positive\ Predictive\ Value \times True\ Positive\ Rate\ \times \\ True\ Negative\ Rate \times Negative\ Predictive\ Value \end{array}} \\ - \sqrt{\begin{array}{c} False\ Discovery\ Rate \times False\ Negative\ Rate\ \times \\ False\ Positive\ Rate \times False\ Omission\ Rate \end{array}}. \tag{18}$$

*Matthews correlation* is a combination of *informedness* and *markedness*. The former considers how informed the classifier's decision with knowledge is compared to "chance" (*Powers, 2007*). Alternatively, *informedness* is paraphrased as a probability with respect to a real variable rather than the "chance" (*Powers, 2013*). Conversely, *markedness* carries information on how possibly the prediction variable will be marked by the *true* variable

(*Layher, Brosch & Neumann, 2017*). Overall, the *Matthews correlation*, as a geometric mean of *informedness* and *markedness*, shows the correlation between the *prediction* and *true* values. The occurrence of this metric is rare in the literature; nonetheless, some intriguing studies have been conducted, such as those on diet recommendations (*Devi, Bhavithra & Saradha, 2020b*).

## EXPERIMENTAL DESIGN

This section describes how we processed the data for various similarity measurements. The overall algorithmic flow is introduced, including the modifications to the equations. The premising *BNC* values were determined prior to its usage in the subsequent section.

We first focus on the algorithm applied during the simulations. The algorithmic flow can guide any prospective RS scientist to follow the basic steps. The procedure for the proposed test package is summarized in Algorithm 1. The details related to several constant parameters, such as *test item count*, *cross-validation fold*, *neighbor counts*, and *liking threshold*, are presented.

Running the algorithm for any user, five random items were considered. The *k-fold cross-validation* technique was integrated with the implementation of repeated randomized test attempts. In each independent analysis, the folds were shuffled, and the test items were alternated. Regarding reliability, each test attempt was performed multiple times and averaged.

Utilizing the fine-tuned $\varepsilon$ parameter, an increased runtime may occur, especially in bulk tests. Several previous studies have initiated this step size as $\varepsilon = 50$ (*Wang, Vries & Reinders, 2006*), $\varepsilon = 10$ (*Sánchez et al., 2008*; *Liu et al., 2013*; *Huang & Dai, 2015*), or $\varepsilon = 5$ (*Ghazanfar & Prugel-Bennett, 2010*; *Sun et al., 2017*). In addition, *Feng et al. (2018)* measured different similarity measures by setting $\varepsilon = 5$; however, they focused only on the error metrics. *Bag, Kumar & Tiwari (2019)* illustrated the performance of the metrics using discrete BNC values of 5, 20, 50, and 100. Considering our test package, the fine-tuned neighbor step size $\varepsilon$, is set distinguishingly compared to the previous studies. $\varepsilon = 1$ was chosen to monitor the sensitivity of the tests, and the neighboring interval produced smooth findings.

Furthermore, one of the main perspectives of previous efforts is to limit the computation time by generalizing several parameters in the global scope of a development environment. Considering each iteration of the test package, although utilizing globally computed arguments reduces the runtime of experiments, it lacks the necessary dynamic perspective for real-time application imitation. Therefore, we performed Algorithm 1 for all similarity equations to examine this fallacy.

We applied the algorithm to two separate datasets to investigate the overall performance. A fine-tuned neighboring approach was performed for the best combination of the *similarity equation* and *performance metrics*. After selecting test items throughout steps one to three, all possible similarity equations were called in step four. A clear picture of the equations and corresponding modifications for dynamicity and weight significance are presented in Table 7. A parametric neighborhood was applied from 1 to 100 users with a single increment as shown in step five. Regarding each loop iteration in

| Algorithm 1 | Pseudocode of the experimental process for an individual test package. |
|---|---|
| A × I | The size of the dataset, A is for users (row count), |
| | I is for items (column count). |
| R = 5 | Randomly selected test items, $\binom{I_a}{5}$ for each user. |
| $k$ = 10 | 10-fold cross-validation. |
| LNC = 1 | BNC minimum value parameter. |
| MNC = 100 | BNC maximum value parameter. |
| $\varepsilon$ = 1 | Fine-tuned BNC increment parameter. |
| $t$ = 3.5 | Binary prediction (*liked* or *disliked*) rating threshold (for 5-star scale). |

1.    Create test *ItemSet* (A×R) randomly and set *k*-fold parameters

2.    **for** all users *a* = 1:A associated with *k*-folds

    3.  **for** all items *i* = 1:R in the corresponding row of *ItemSet*

        4.  **for each** *SimEq*

            5.  **for** all *bnc* = LNC : $\varepsilon$ : MNC

                6.  $BN \leftarrow getBestNeighbors(SimEq, a, i, bnc)$;

                *// for (a,i) pair, using the Train Set of corresponding folds*

                7.  $p_{a,i}^{SimEq,bnc} \leftarrow calculatePrediction(BN)$;

                **endfor**

            **endfor**

        **endfor**

    **endfor**

**for** all *a*, *i*, *SimEq*, *bnc*

8.  *evaluatePerformance*($p_{a,i}^{SimEq,bnc}$, *t*);

*// exact and threshold performance analysis for all $p_{a,i}^{SimEq,bnc}$*

  **endfor**

step six, the best neighbors were selected based on the similarity score, which was sorted for all neighboring nominees, indicating the users who rated the test item. After the prediction calculation in step seven, the performance was evaluated in step eight.

During the correlation computations, we emphasize the possible shortcomings of readily available functions on computing platforms. Correlation methods are mostly inline functions in a development environment. However, we strongly suggest checking the built-in functions for statistical parameter calculation, such as the *mean* and *median*. It is advised not to consider the statistics of only the co-rated items during the computations. It is more accurate to include the statistics of all items in analyzing general behavior and shared characteristics, especially in the context of dynamic RS (*Okyay & Aygun, 2021*).

## RESULTS AND DISCUSSION

In this section, several repeated randomized tests were analyzed considering various performance metrics. The preliminary findings related to the best-performing *BNC*s are

**Table 7  All test configurations considering nIOI and SW similarity measurements.**

| Abbreviation of similarity equation | Dynamic | Significance weighting | Related equation | BNC LNC : ε : MNC |
|---|---|---|---|---|
| $PCC$ | | | Eq. (1) | 1:1:100 |
| $PCC^{sw}$ | | ✓ | Eq. (7) | 1:1:100 |
| $PCC_{nIOI}$ | ✓ | | Eq. (6) | 1:1:100 |
| $PCC^{sw}_{nIOI}$ | ✓ | ✓ | Eq. (14) | 1:1:100 |
| $MRC$ | | | Eq. (2) | 1:1:100 |
| $MRC^{sw}$ | | ✓ | Eq. (9) | 1:1:100 |
| $MRC_{nIOI}$ | ✓ | | Eq. (8) | 1:1:100 |
| $MRC^{sw}_{nIOI}$ | ✓ | ✓ | Eq. (14) | 1:1:100 |
| $COS$ | | | Eq. (3) | 1:1:100 |
| $COS^{sw}$ | | ✓ | Eq. (11) | 1:1:100 |
| $COS_{nIOI}$ | ✓ | | Eq. (10) | 1:1:100 |
| $COS^{sw}_{nIOI}$ | ✓ | ✓ | Eq. (14) | 1:1:100 |
| $JAC$ | | | Eq. (4) | 1:1:100 |
| $JAC^{sw}$ | | ✓ | Eq. (13) | 1:1:100 |
| $JAC_{nIOI}$ | ✓ | | Eq. (12) | 1:1:100 |
| $JAC^{sw}_{nIOI}$ | ✓ | ✓ | Eq. (14) | 1:1:100 |

first given to provide guide to the subsequent sections. Then, the effect of sensitive neighboring interval selection for different similarity equations is performed, thereby measuring the importance of dynamicity and weight significance.

First, the best-performing *BNC* values of the related performance monitoring are discovered considering each similarity measure in Table 7. After setting *BNC* precisely, the observations under the various performance metrics are summarized in Tables 8 and 9. Table 8 shows the ML100K-based *BNC* values recorded for the dynamicity and weight significance approaches. Meanwhile, Table 9 presents the same analyses of the ML1M. These dataset-oriented analyses facilitate guidance for further metric comparisons in the subsequent section. Here, *BNC* values inspected beforehand are utilized to interpret the adequate weight-metric combinations.

Moreover, the preliminary results presented in Tables 8 and 9 highlight the effect of the *SW* method in terms of *BNC*. For instance, *PCC* benefits the reduced *BNCs* with the best performance when *SW* is applied. Excluding *specificity* and *fallout* in the ML100K, *PCC* has the advantage of the *SW* method. As presented in Table 8, $PCC^{sw}_{nIOI}$ achieves the top performance when *BNC* = 17 for *markedness*, *Matthews correlation*, *F1-measure*, threshold-based error metrics and *accuracy*, *Fowlkes–Mallows index*, *threat score*, *inverse precision*, *sensitivity*, *miss rate*, and *false omission rate* in the ML100K. Similarly, considering the ML1M, the same observation with *BNC* = 31 is valid for *markedness*, *Matthews correlation*, *F1-measure*, threshold-based error metrics, *exact accuracy*, *threshold accuracy*, *Fowlkes-Mallows index*, *threat score*, *inverse precision*, *sensitivity*, *miss rate*, and *false omission rate*. Further, the same *BNC* monitoring in terms of *MRC* was performed. *F1-measure*, *exact MAE*, *exact accuracy*, *Fowlkes-Mallows index*, *threat score*,

**Table 8** The best-performing BNC values under a variety of performance metrics: ML100K.

| Equation | Performance metrics | | | | | | | | | | | | | | | | | | | | | | | |
|---|---|---|---|---|---|---|---|---|---|---|---|---|---|---|---|---|---|---|---|---|---|---|---|---|
| | *Markedness* | *Informedness* | *Matthews correlation* | *F1-measure* | *MAE exact* | *MSE exact* | *RMSE exact* | *MAE threshold* | *MSE threshold* | *RMSE threshold* | *Accuracy exact* | *Accuracy threshold* | *Accuracy balanced* | *Fowlkes-Mallows index* | *Prevalence threshold* | *Threat score* | *Precision* | *Inverse precision* | *Sensitivity/Recall* | *Specificity* | *Fallout* | *Miss rate* | *False discovery rate* | *False omission rate* |
| $PCC_{nIOI}$ | 45 | 34 | 45 | 47 | 34 | 23 | 23 | 45 | 45 | 45 | 34 | 45 | 34 | 47 | 34 | 47 | 34 | 47 | 100 | 17 | 17 | 100 | 34 | 47 |
| $PCC_{nIOI}^{sw}$ | 17 | 24 | 17 | 17 | 24 | 22 | 22 | 17 | 17 | 17 | 26 | 17 | 24 | 17 | 26 | 17 | 26 | 17 | 17 | 53 | 53 | 17 | 26 | 17 |
| ↓/↑ | ↓ | ↓ | ↓ | ↓ | ↓ | ↓ | ↓ | ↓ | ↓ | ↓ | ↓ | ↓ | ↓ | ↓ | ↓ | ↓ | ↓ | ↓ | ↓ | ↑ | ↑ | ↓ | ↓ | ↓ |
| $MRC_{nIOI}$ | 26 | 26 | 26 | 42 | 27 | 19 | 19 | 26 | 26 | 26 | 27 | 26 | 26 | 42 | 25 | 42 | 25 | 42 | 59 | 12 | 12 | 59 | 25 | 42 |
| $MRC_{nIOI}^{sw}$ | 27 | 28 | 28 | 27 | 23 | 20 | 20 | 27 | 27 | 27 | 23 | 27 | 28 | 27 | 30 | 27 | 30 | 27 | 17 | 37 | 37 | 17 | 30 | 27 |
| ↓/↑ | ↑ | ↑ | ↑ | ↓ | ↓ | ↑ | ↑ | ↑ | ↑ | ↑ | ↓ | ↑ | ↑ | ↓ | ↑ | ↓ | ↑ | ↓ | ↓ | ↑ | ↑ | ↓ | ↑ | ↓ |
| $COS_{nIOI}$ | 27 | 27 | 27 | 25 | 28 | 35 | 35 | 27 | 27 | 27 | 27 | 27 | 27 | 25 | 36 | 25 | 36 | 25 | 25 | 100 | 100 | 25 | 36 | 25 |
| $COS_{nIOI}^{sw}$ | 31 | 50 | 50 | 18 | 50 | 67 | 67 | 31 | 31 | 31 | 24 | 31 | 50 | 14 | 99 | 18 | 99 | 14 | 9 | 100 | 100 | 9 | 99 | 14 |
| ↓/↑ | ↑ | ↑ | ↑ | ↓ | ↑ | ↑ | ↑ | ↑ | ↑ | ↑ | ↓ | ↑ | ↑ | ↓ | ↑ | ↓ | ↑ | ↓ | ↓ | ⇔ | ⇔ | ↓ | ↑ | ↓ |
| $JAC_{nIOI}$ | 39 | 39 | 39 | 39 | 40 | 40 | 40 | 39 | 39 | 39 | 36 | 39 | 39 | 39 | 39 | 39 | 39 | 39 | 35 | 20 | 20 | 35 | 39 | 39 |
| $JAC_{nIOI}^{sw}$ | 32 | 45 | 45 | 29 | 45 | 59 | 59 | 36 | 36 | 36 | 44 | 36 | 45 | 29 | 100 | 29 | 100 | 29 | 18 | 100 | 100 | 18 | 100 | 29 |
| ↓/↑ | ↓ | ↑ | ↑ | ↓ | ↑ | ↑ | ↑ | ↓ | ↓ | ↓ | ↑ | ↓ | ↑ | ↓ | ↑ | ↓ | ↑ | ↓ | ↓ | ↑ | ↑ | ↓ | ↑ | ↓ |

**Notes:**
↓ : The best-performing BNC value reduces *via SW*.
↑ : The best-performing BNC value increases *via SW*.

*inverse precision*, *sensitivity*, *miss rate*, and *false omission rate* benefit from *SW* in terms of achieving lower *BNCs* in the ML100K. Half of the metrics showed their top performance for *BNC* = 27 and 28. In contrast, relatively higher *BNC* values are required in the ML1M for the top performances. More than half of the metrics work well for *BNCs* 23 and 44 when the *SW* approach is applied. The performance of *COS* in the ML100K in terms of the metrics that perform well with regard to lower *BNCs* is similar to *MRC*. These are *F1-measure*, *exact accuracy*, *Fowlkes–Mallows index*, *threat score*, *inverse precision*, *sensitivity*, *miss rate*, and *false omission rate*. In the ML1M, *COS* is the least effective similarity equation in terms of the numbers of metrics, which benefit from *SW* reducing the *BNCs*. However, *JAC* in the ML1M leads all other equations by having almost all metrics (except for *informedness*, *exact MSE*, and *RMSE*) performing well concerning lower *BNCs*. Overall, the *SW* approach is compatible with *F1-measure*, *Fowlkes–Mallows index*, *threat score*, *inverse precision*, *sensitivity*, *miss rate*, and *false omission rate*. This indicates lower *BNCs* when *SW* is applied for all touchstone similarity equations both in the ML100K and ML1M. These observations can be visually inferred from Tables 8 and 9. In the following subsections, the evaluations related to the analyses of overall metrics, including the hybrid monitoring to achieve the adequate weight-metric combination, are discussed.

**Table 9 The best-performing BNC values under a variety of performance metrics: ML1M.**

| Equation | Markedness | Informedness | Matthews correlation | F1-measure | MAE exact | MSE exact | RMSE exact | MAE threshold | MSE threshold | RMSE threshold | Accuracy exact | Accuracy threshold | Accuracy balanced | Fowlkes-Mallows index | Prevalence threshold | Threat score | Precision | Inverse precision | Sensitivity/Recall | Specificity | Fallout | Miss rate | False discovery rate | False omission rate |
|---|---|---|---|---|---|---|---|---|---|---|---|---|---|---|---|---|---|---|---|---|---|---|---|---|
| $PCC_{nIOI}$ | 100 | 91 | 91 | 100 | 99 | 91 | 91 | 100 | 100 | 100 | 96 | 100 | 91 | 100 | 31 | 100 | 31 | 100 | 100 | 4 | 4 | 100 | 31 | 100 |
| $PCC_{nIOI}^{sw}$ | 31 | 93 | 31 | 31 | 100 | 100 | 100 | 31 | 31 | 31 | 31 | 31 | 93 | 31 | 100 | 31 | 100 | 31 | 31 | 100 | 100 | 31 | 100 | 31 |
| ↓/↑ | ↓ | ↑ | ↓ | ↓ | ↑ | ↑ | ↑ | ↓ | ↓ | ↓ | ↓ | ↓ | ↑ | ↓ | ↑ | ↓ | ↑ | ↓ | ↓ | ↑ | ↑ | ↓ | ↑ | ↓ |
| $MRC_{nIOI}$ | 97 | 92 | 92 | 97 | 97 | 60 | 60 | 92 | 92 | 92 | 92 | 92 | 92 | 97 | 48 | 97 | 48 | 97 | 100 | 8 | 8 | 100 | 48 | 97 |
| $MRC_{nIOI}^{sw}$ | 44 | 93 | 44 | 44 | 58 | 94 | 94 | 44 | 44 | 44 | 41 | 44 | 93 | 44 | 93 | 44 | 93 | 23 | 23 | 96 | 96 | 23 | 93 | 23 |
| ↓/↑ | ↓ | ↑ | ↓ | ↓ | ↓ | ↑ | ↑ | ↓ | ↓ | ↓ | ↓ | ↓ | ↑ | ↓ | ↑ | ↓ | ↑ | ↓ | ↓ | ↑ | ↑ | ↓ | ↑ | ↓ |
| $COS_{nIOI}$ | 41 | 36 | 41 | 41 | 45 | 45 | 45 | 41 | 41 | 41 | 72 | 41 | 36 | 50 | 13 | 41 | 13 | 50 | 83 | 4 | 4 | 83 | 13 | 50 |
| $COS_{nIOI}^{sw}$ | 30 | 92 | 72 | 30 | 97 | 100 | 100 | 72 | 72 | 72 | 91 | 72 | 92 | 30 | 99 | 30 | 99 | 30 | 18 | 99 | 99 | 18 | 99 | 30 |
| ↓/↑ | ↓ | ↑ | ↑ | ↓ | ↑ | ↑ | ↑ | ↑ | ↑ | ↑ | ↑ | ↑ | ↑ | ↓ | ↑ | ↓ | ↑ | ↓ | ↓ | ↑ | ↑ | ↓ | ↑ | ↓ |
| $JAC_{nIOI}$ | 58 | 29 | 58 | 84 | 58 | 53 | 53 | 58 | 58 | 58 | 58 | 58 | 29 | 84 | 29 | 84 | 29 | 90 | 99 | 10 | 10 | 99 | 29 | 90 |
| $JAC_{nIOI}^{sw}$ | 54 | 56 | 56 | 54 | 55 | 55 | 55 | 56 | 56 | 56 | 54 | 56 | 56 | 75 | 25 | 54 | 25 | 75 | 75 | 4 | 4 | 75 | 25 | 75 |
| ↓/↑ | ↓ | ↑ | ↓ | ↓ | ↓ | ↑ | ↑ | ↓ | ↓ | ↓ | ↓ | ↓ | ↑ | ↓ | ↓ | ↓ | ↓ | ↓ | ↓ | ↓ | ↓ | ↓ | ↓ | ↓ |

**Notes:**
↓ : The best-performing *BNC* value reduces *via SW*.
↑ : The best-performing *BNC* value increases *via SW*.

## A. Analyses of the preeminent metrics

First, the preeminent metrics (such as *informedness*, *markedness*, and *Matthews correlation*) and the *F1-measure* are utilized to show the comparative performance plots of all the similarity equations in Table 7 using each individual metric. The ML100K and ML1M releases were analyzed separately. Considering the plots, the *x*- and *y-axis* represent the fine-tuned *BNCs* and related metric output, respectively.

The statistical approach is depicted in this study to set a dynamic environment which requires a more adaptive procedure. The results, based on hypothetical computations, can only determine the maximum achievable top-performance of the dynamicity concept. We prove that dynamicity deviates from the maximum reachable results. In the subsequent figures, the dashed lines represent the theoretical perspective by including the global-only statistics, causing a fallacy. Moreover, the solid lines represent dynamicity with the *nIOI* approach. In addition, lines with diamond marks illustrate the results free from the *SW* approach, whereas the *SW* adjustment can be monitored through the unmarked lines.

Considering Figs. 4 and 5, performance plots of ML100K and ML1M are provided for the preeminent metrics and *F1-measure*. Each row of subplots was compared to an equation-dependent perspective. Analyzing the ML100K, the similarity equation with only the *SW* modification achieves the best results for the *PCC* lines in black. Regarding the

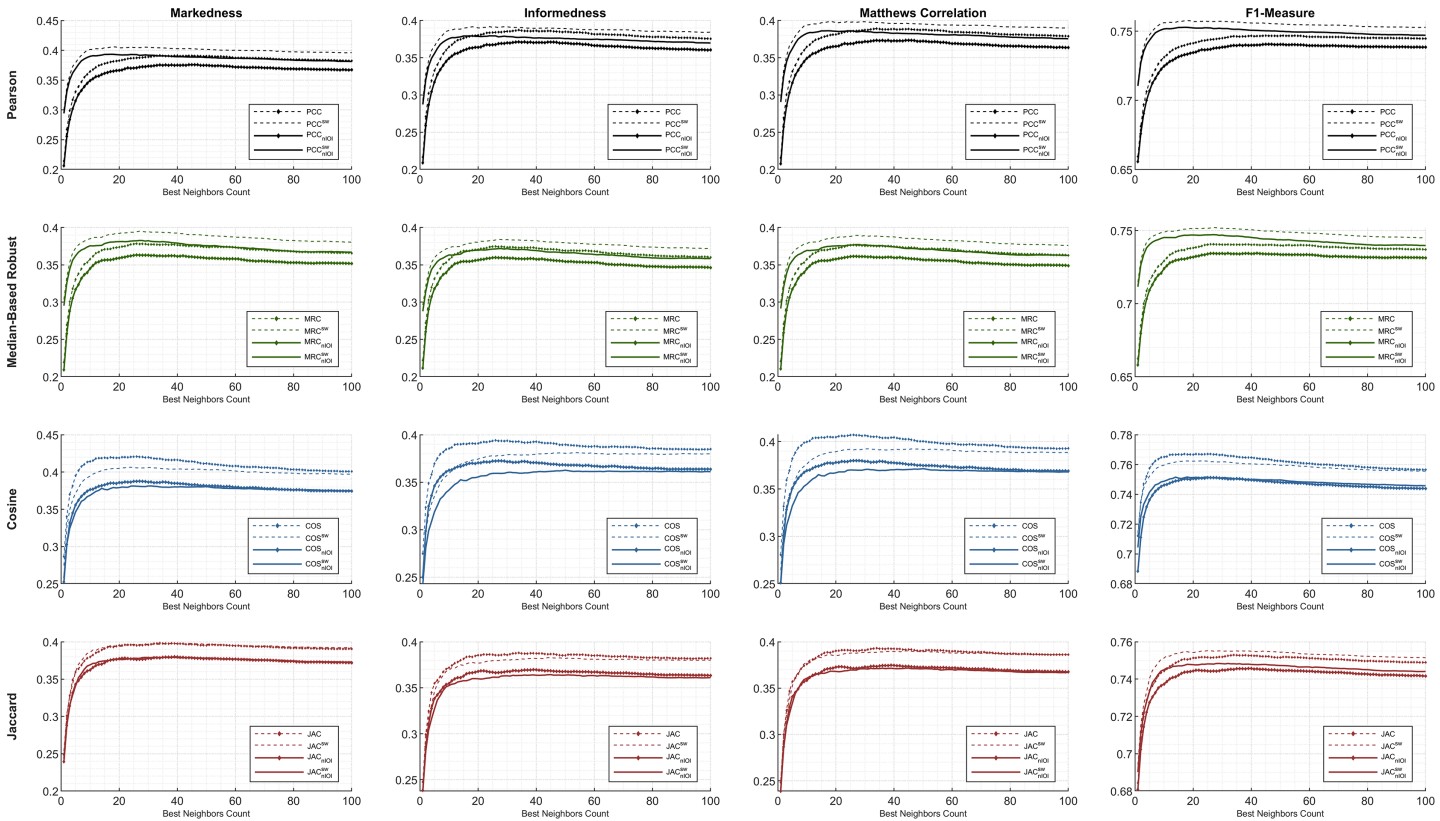

**Figure 4 The evaluation over ML100K: similarity weight and preeminent metric combination to compare the dynamicity and weight significance.**

*MRC* (the lines with a green color), the same dominance for the *SW* methodology may be observed through all the metrics. Although *SW* does not boost the performance of the *COS* in all the metrics, plots with *SW* in *JAC* show similar performance compared to those without *SW*.

Comparatively, we present the same metric performance throughout the similarity measures in the context of ML1M as shown in Fig. 5. Including only the dynamic *COS*, all other similarities with *SW* increase the *F1-measure* performance. However, the performance in *informedness* diminishes compared to that in Fig. 4. The top- and least-performing lines in each similarity measure remain the same for *markedness*, *Matthews correlation*, and *F1-measure* as in the ML100K. Nonetheless, regarding *informedness*, the effect of the *SW* in the *PCC* and *MRC* interchanges the least-performing similarity equation. Furthermore, dynamicity resulted in the same expected outcomes in the ML1M analysis.

## B. Hybrid monitoring considering the preeminent metrics

This subsection depicts the overall comparison of the compelled dynamicity with the applied weight significance. Considering Fig. 6A, for ML100K, *PCC* is notably in the leading position compared to other similarity measurements. The ranking for the rest of

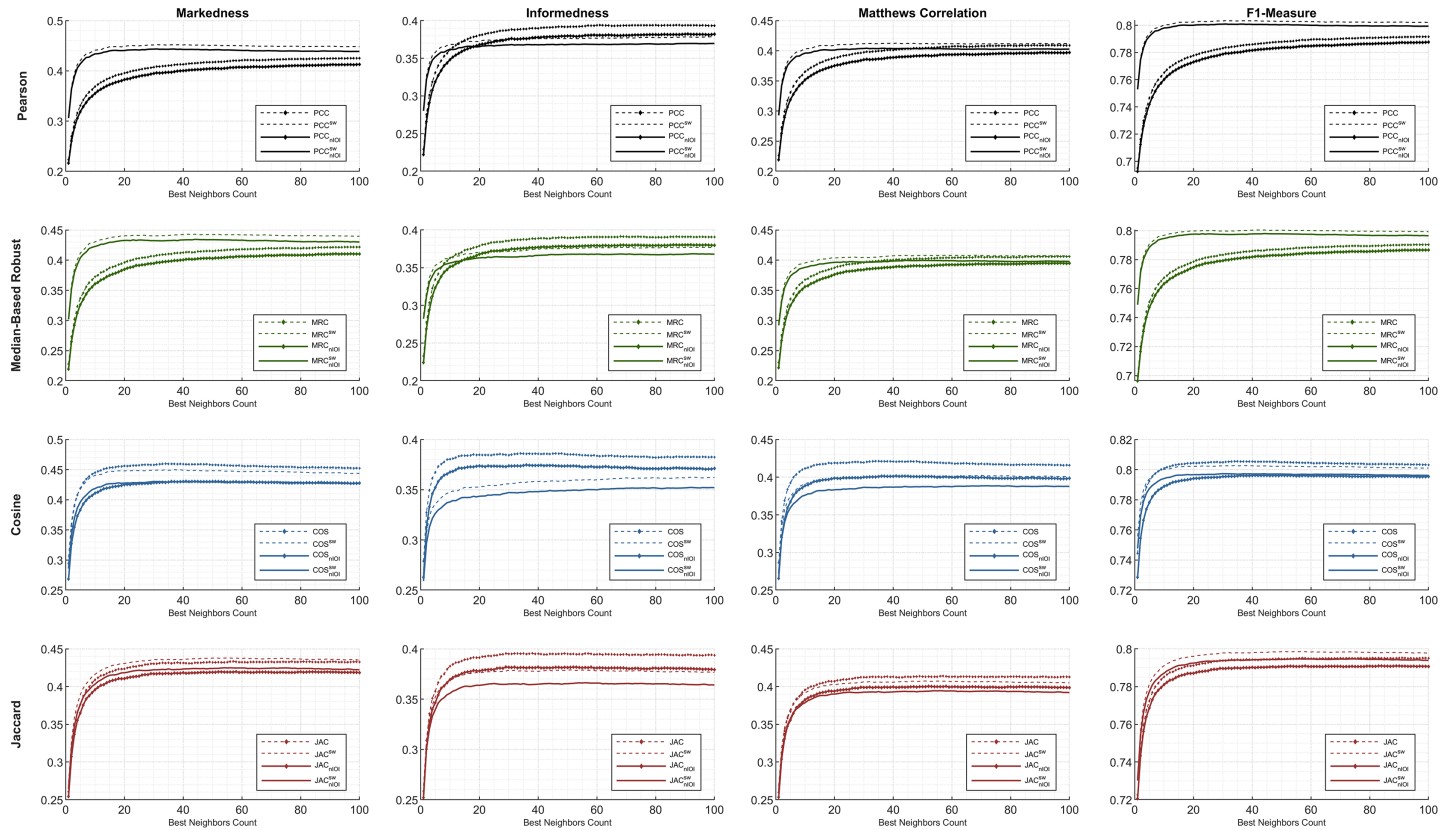

**Figure 5 The evaluation over ML1M: similarity weight and preeminent metric combination to compare the dynamicity and weight significance.**

the similarity measures is difficult to generalize because the lines interchangeably depend on the *BNC*. Regarding *markedness*, *MRC* starts with better performance for fewer *BNCs*; however, the trend reverses for *BNC* > 27. In addition, measurements other than *MRC* performed better for *BNC* > 40. A similar behavior is valid for the *informedness* and *Matthews correlation*; nevertheless, each of them has a relatively greater *BNC* threshold. Approximately, *BNC* > 75 for *informedness* and *BNC* > 60 for *Matthews correlation* decayed the performance of the *MRC*. Considering the *F1-measure*, a relatively stable performance was obtained for the measurements when *BNC* > 10. Ranking the performance of the equations, the *F1-measure* can be considered as the most stable metric independent of the *BNC* for the ML100K. This makes it appropriate comparisons of similarity measure performances without further *BNC* considerations.

Regarding Fig. 6B, the same hybrid monitoring of *nIOI* and *SW* is presented for the ML1M. The top-performing lines of the abovementioned preeminent metrics were still obtained *via* the *PCC*. The *COS*, compared to the others, has a relatively poor performance for the *informedness* metric. The performance ranking of the similarity equations remains more stable as a function of the *BNC* compared to the ML100K. There are slight interchanges between *MRC* and *JAC* only in *informedness*. Nonetheless, considering the

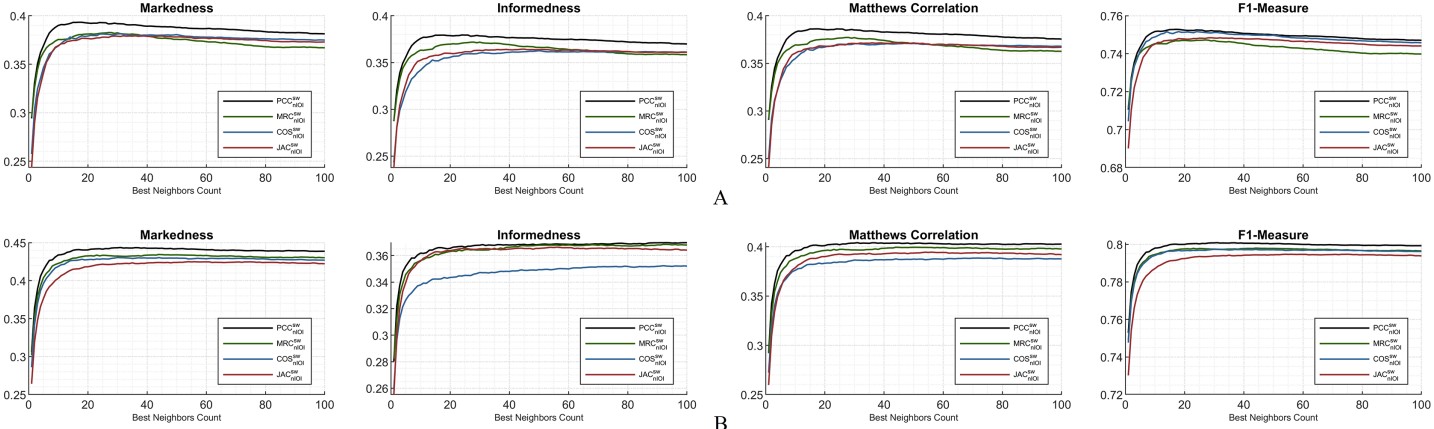

**Figure 6 Markedness, informedness, and Matthews correlation as preeminent metrics, plus F1-measure highlighting the hybrid monitoring for (A) ML100K and (B) ML1M.**

others, the relative performance is independent of the *BNC*. Contrary to the significant interchanges in the ML100K, performance metrics maintain their relative positions in the ML1M. This stability finding can be a general interpretation, and it can be concluded that the larger the dataset is, the more stable the performance.

## C. Extensive analysis of other metrics for hybrid monitoring

The subsequent figures illustrate the extensive analysis of other metrics frequently evaluated in the literature. First, the ML100K plots with hybrid monitoring are presented in Fig. 7.

Regarding accuracy-based metrics, both the *exact* (sensitive rating prediction) and *binary* (*liked* or *disliked* labeling) performances were monitored. The *PCC* had a relatively higher accuracy margin of approximately 0.05 for both metrics. Another accuracy calculation is performed extensively in this study. Considering *balanced accuracy*, the *PCC* outperformed for all the *BNCs*, whereas the *MRC* diminished. Similarly, the error metrics were measured, considering both *exact* and *binary* performances. As expected, the *binary prediction error* rates were lower than the *exact prediction error* rates. Both are plotted using *MAE*, *MSE*, and *RMSE*. Considering the *binary prediction error*, the *PCC* achieved the lowest possible error rates. Nonetheless, regarding the *exact accuracy* metric, the top performance for low error rates was interchangeable based on the neighborhood. Approximately *BNC* > 40, *BNC* > 20, and *BNC* > 20 for *MSE*, *MAE*, and *RMSE*, respectively, yielded lower error rates with the *JAC* measure.

In addition, the *Fowlkes–Mallows index* shows that *PCC* and *COS* achieve outstanding performances, whereas in general, the *MRC* has poor performance. Considering the *threat score*, the hits of user dislikes were not included; however, the ratio of liking matches concerning the misses is checked. Similarity measurement rankings are relatively stable after *BNC* = 10 and *PCC* was an adequate measure, while *MRC* fell behind. This implies that, after the *BNC* value of 10, the ranking of the metric values from similarity equations relative to the *y*-axis remains stable.

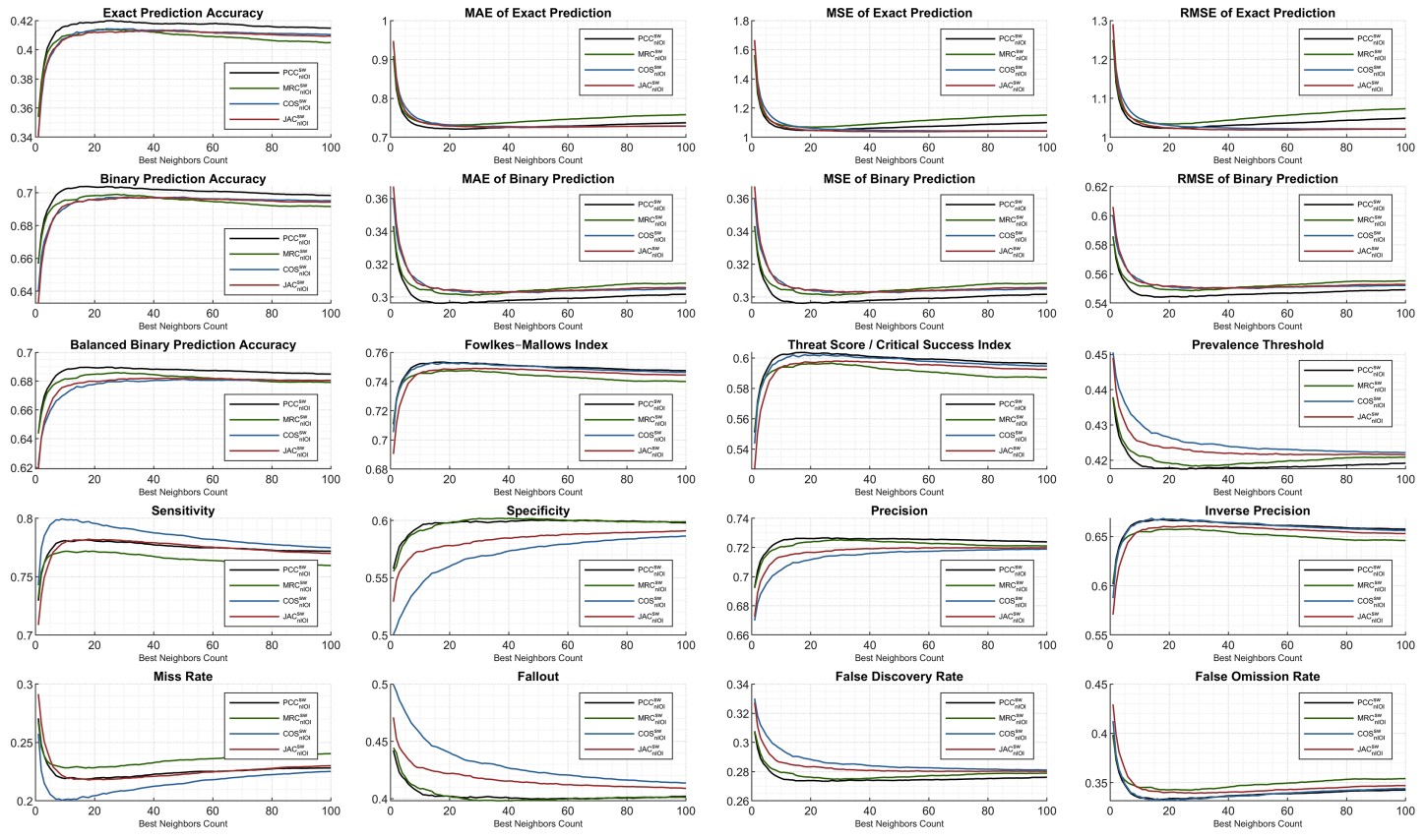

**Figure 7** Extended performance evaluation on ML100K considering the hybrid monitoring.

This study also discusses different performance checks based on the interdisciplinary applications discussed previously. Considering the context of RS, we propose the application of *a prevalence threshold* that sources compound information, including several associated metrics inferred as $\left(\sqrt{sensitivity \times fallout} - fallout\right)/informedness$. Similar to the error metrics, the lesser the *prevalence threshold* value is, the more exactitude accomplished. Although higher values are obtained in the *COS* compared to the others, the *PCC* has proven its superiority with lower rates.

Some metrics, such as *sensitivity* and *specificity*, provide information on how likely the top-*n* items match the user's taste or vice versa. Correctly identified positives (*i.e.*, *sensitivity*) perform well for the lower *BNC*s, and the *COS* measure leads with a peak of approximately *BNC* = 9. Correctly identified negatives (*i.e.*, *specificity*) are distinctively better as the neighbor count increases for *COS* and *JAC*, whereas *PCC* and *MRC* are relatively stable.

The last two rows of the subplots are complementary metric couples. The metrics of *sensitivity* and *miss rate*; *specificity* and *fallout*; *precision* and *false discovery rate*; *inverse precision* and *false omission rate* complete each other. Considering *specificity*, *precision*, and *inverse precision*, *PCC* performs adequately, which can be verified from *fallout*, *false discovery rate*, and *false omission rate*.

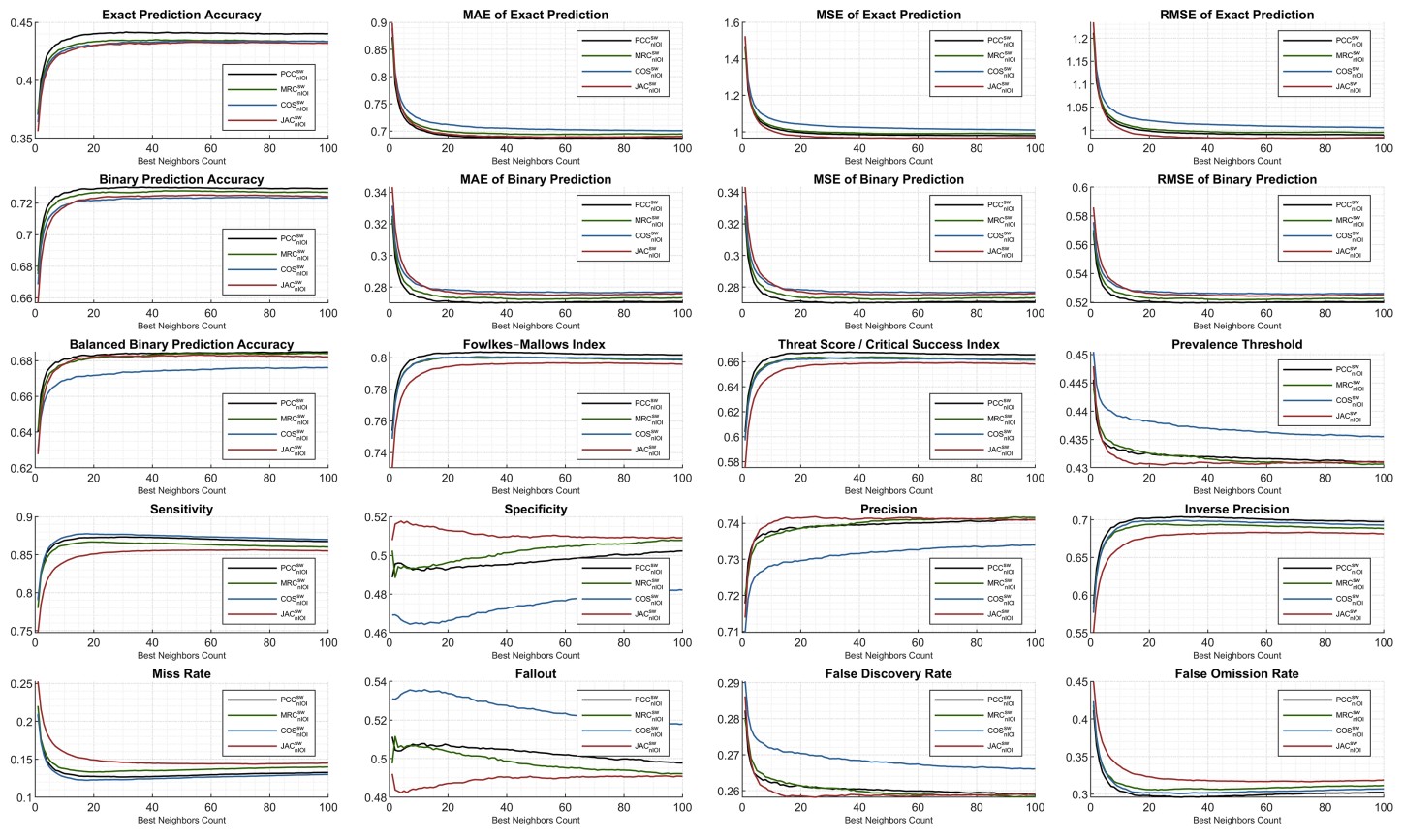

**Figure 8 Extended performance evaluation on ML1M considering the hybrid monitoring.**

The ML1M evaluation is presented comparatively to the previous findings of the ML100K. Regarding Fig. 8, the analyses are illustrated for the same evaluated metrics. The trend in the *exact accuracy* was similar to that of the ML100K with greater scores. The *PCC* puts a margin, whereas the others perform closer to each other with slightly lower values compared to the *PCC*. In addition, the *exact prediction error* metrics generally result in a reduced numerical range. The most erroneous metric is the *COS*, which is valid for both *exact* and *binary prediction error* metrics. The error performance in the ML1M is relatively stable, and the *PCC* is still a less fragile metric for *binary prediction*. Considering *binary* analyses, similarity measures are homogenously ranked again with the dominance of the *PCC*.

Furthermore, the *Fowlkes–Mallows index* shows an increased range of scoring with respect to the ML100K findings. Regarding the *threat score*, it was observed that the *MRC* significantly improved compared to the others. Considering the *prevalence threshold*, *COS* had a higher margin than the others compared to the performances in the previous analysis. Moreover, although the *COS* has a good *sensitivity* observation, it performs worse in terms of *specificity* and *precision*, as demonstrated in the previous findings. This indicates that the *COS* suffers from *a true negative rate* and *positive predictive value*.

The monitoring of smooth *sensitivity* in the ML1M is an important feedback because it is a component of some compound metrics. On the contrary, an indicative finding from the comparison of both releases is the behavior of the *specificity* and *fallout* metric couples. Considering the ML100K, a relatively more stable distribution is monitored for increasing the *BNC* values, whereas in the ML1M, the behavior becomes unstable. Lastly, whereas *JAC* for *precision* increases in the ranking compared to the previous findings, the opposite is valid for the *inverse precision*.

### D. Adequate weight-metric combination of the top-performing BNCs

Having represented all the plots, we summarize the test results using a tabular structure for a compact heat-map presentation. Considering Tables 10 and 11, the performance metrics for each similarity measurement are visualized in a colored format[2]. Considering the preliminarily explanations in the third section, the selected *BNC*s achieving the top performances are added to the tables, highlighting the main motivation of our study: *the decision of adequate weight-metric combinations*. Each metric is processed through column-wise coloring to make the comparison easier. Therefore, each coloring is evaluated within its own column. This indicates that the same color may correspond to different values in other columns; nevertheless, only a single column should be considered to interpret the coloring for any metric. The comparison of the similarity methods in the vertical direction is targeted, considering the neighborhoods. At the end of each heat-map table, the minimum and maximum values referenced in the coloring of the relevant column are shown. The tables demonstrate the comparison by addressing the different correlation equations over the outstanding neighborhoods; thereby, comparing approaches such as dynamicity and *SW*, considering each independent metric. The cells shaded in green indicate the effectiveness of the appropriate combination. We present the results using both the *SW*-induced dynamic equations and plain dynamicity; hence, the effect of weight boosting is monitored.

All the other test outcomes are found in our code repository[3]. We have prepared a fully detailed supplemental material to include all the outcomes. Any RS researcher can benefit from the prepared document for such purposes (*e.g.*, the selection of *BNC*s, enhanced similarity measure conditions, *etc.*), and every iteration in the test package has been logged into the abovementioned document.

The colorized tables are organized by grouping the column names attributed to the performance metrics. The first group is the prioritized *preeminent metrics* in this study. Subsequently, the *error-based metrics* are combined. The third group is *accuracy-based metrics*, and the final group is the rest of the metrics, which are frequently used in the literature, including interdisciplinary applications. This method of representation determines the consistency of each similarity equation, considering the groupings. Furthermore, because the tables are multi-dimensional, they include metrics, correlation methods, and multiple parameters, such as *BNC*, dynamicity, and *SW*. Because the neighborhood calculation makes the tests dependent on a parameter in recommendation systems, the performance of the correlation is better if it is less dependent on the neighboring users. Therefore, column-wise homogeneity indicates less dependence on the

[2] The fractional values in the table are displayed based on three significant digits. The heat-map coloring is achieved according to full precision.

[3] The supplementary material containing the complete results of the whole test package can be accessed from the repository given in the Acknowledgments.

Okyay and Aygun (2021), PeerJ Comput. Sci., DOI 10.7717/peerj-cs.784

**Table 10 Adequate weight-metric combination of the top-performing BNCs: ML100K.**

| Equation | BNC | Markedness | Informedness | Mathews correlation | F1-measure | MAE exact | MSE exact | RMSE exact | MAE threshold | MSE threshold | RMSE threshold | Accuracy exact | Accuracy threshold | Accuracy balanced | Fowlkes-Mallows index | Prevalence threshold | Threat score | Precision | Inverse precision | Sensitivity/Recall | Specificity | Fallout | Miss rate | False discovery rate | False omission rate |
|---|---|---|---|---|---|---|---|---|---|---|---|---|---|---|---|---|---|---|---|---|---|---|---|---|---|
| $PCC_{nIOI}$ | 17 | 0.364 | 0.363 | 0.364 | 0.733 | 0.744 | 1.081 | 1.040 | 0.310 | 0.310 | 0.557 | 0.403 | 0.690 | 0.681 | 0.733 | 0.416 | 0.579 | 0.729 | 0.635 | 0.737 | 0.626 | 0.374 | 0.263 | 0.271 | 0.365 |
| | 23 | 0.371 | 0.368 | 0.369 | 0.737 | 0.740 | 1.075 | 1.037 | 0.307 | 0.307 | 0.554 | 0.405 | 0.693 | 0.684 | 0.737 | 0.416 | 0.583 | 0.730 | 0.641 | 0.744 | 0.624 | 0.376 | 0.256 | 0.270 | 0.359 |
| | 34 | 0.375 | 0.371 | 0.373 | 0.740 | 0.739 | 1.078 | 1.038 | 0.304 | 0.304 | 0.552 | 0.407 | 0.696 | 0.686 | 0.740 | 0.415 | 0.587 | 0.730 | 0.645 | 0.750 | 0.622 | 0.378 | 0.250 | 0.270 | 0.355 |
| | 45 | 0.376 | 0.371 | 0.374 | 0.741 | 0.742 | 1.091 | 1.044 | 0.304 | 0.304 | 0.551 | 0.406 | 0.696 | 0.686 | 0.741 | 0.416 | 0.588 | 0.729 | 0.647 | 0.752 | 0.619 | 0.381 | 0.248 | 0.271 | 0.353 |
| | 47 | 0.376 | 0.371 | 0.373 | 0.741 | 0.742 | 1.092 | 1.045 | 0.304 | 0.304 | 0.552 | 0.406 | 0.696 | 0.685 | 0.741 | 0.416 | 0.588 | 0.729 | 0.647 | 0.753 | 0.618 | 0.382 | 0.247 | 0.271 | 0.353 |
| | 100 | 0.367 | 0.360 | 0.364 | 0.739 | 0.756 | 1.141 | 1.068 | 0.308 | 0.308 | 0.555 | 0.403 | 0.692 | 0.680 | 0.739 | 0.419 | 0.586 | 0.723 | 0.644 | 0.754 | 0.606 | 0.394 | 0.246 | 0.277 | 0.356 |
| $PCC_{nIOI}^{sw}$ | 17 | 0.393 | 0.379 | 0.386 | 0.753 | 0.722 | 1.048 | 1.023 | 0.296 | 0.296 | 0.544 | 0.419 | 0.704 | 0.690 | 0.753 | 0.418 | 0.604 | 0.726 | 0.667 | 0.781 | 0.598 | 0.402 | 0.219 | 0.274 | 0.333 |
| | 22 | 0.392 | 0.379 | 0.385 | 0.752 | 0.721 | 1.046 | 1.023 | 0.297 | 0.297 | 0.545 | 0.419 | 0.703 | 0.689 | 0.753 | 0.418 | 0.603 | 0.726 | 0.666 | 0.780 | 0.598 | 0.402 | 0.220 | 0.274 | 0.334 |
| | 24 | 0.393 | 0.379 | 0.386 | 0.753 | 0.721 | 1.047 | 1.023 | 0.296 | 0.296 | 0.544 | 0.420 | 0.704 | 0.690 | 0.753 | 0.417 | 0.603 | 0.727 | 0.666 | 0.780 | 0.599 | 0.401 | 0.220 | 0.273 | 0.334 |
| | 26 | 0.393 | 0.379 | 0.386 | 0.752 | 0.721 | 1.049 | 1.024 | 0.296 | 0.296 | 0.544 | 0.420 | 0.704 | 0.690 | 0.753 | 0.417 | 0.603 | 0.727 | 0.666 | 0.780 | 0.599 | 0.401 | 0.220 | 0.273 | 0.334 |
| | 53 | 0.388 | 0.376 | 0.382 | 0.750 | 0.728 | 1.070 | 1.034 | 0.299 | 0.299 | 0.547 | 0.418 | 0.701 | 0.688 | 0.750 | 0.418 | 0.600 | 0.726 | 0.662 | 0.775 | 0.600 | 0.400 | 0.225 | 0.274 | 0.338 |
| $MRC_{nIOI}$ | 12 | 0.353 | 0.351 | 0.352 | 0.728 | 0.755 | 1.105 | 1.051 | 0.316 | 0.316 | 0.562 | 0.397 | 0.684 | 0.676 | 0.728 | 0.419 | 0.572 | 0.724 | 0.628 | 0.732 | 0.620 | 0.380 | 0.268 | 0.276 | 0.372 |
| | 19 | 0.358 | 0.356 | 0.357 | 0.732 | 0.750 | 1.096 | 1.047 | 0.313 | 0.313 | 0.559 | 0.400 | 0.687 | 0.678 | 0.732 | 0.419 | 0.577 | 0.725 | 0.634 | 0.738 | 0.617 | 0.383 | 0.262 | 0.275 | 0.366 |
| | 25 | 0.363 | 0.360 | 0.361 | 0.734 | 0.748 | 1.098 | 1.048 | 0.311 | 0.311 | 0.557 | 0.402 | 0.689 | 0.680 | 0.734 | 0.418 | 0.580 | 0.726 | 0.637 | 0.743 | 0.617 | 0.383 | 0.257 | 0.274 | 0.363 |
| | 26 | 0.363 | 0.360 | 0.361 | 0.734 | 0.748 | 1.099 | 1.048 | 0.310 | 0.310 | 0.557 | 0.402 | 0.690 | 0.680 | 0.734 | 0.418 | 0.580 | 0.726 | 0.637 | 0.743 | 0.617 | 0.383 | 0.257 | 0.274 | 0.363 |
| | 27 | 0.363 | 0.360 | 0.361 | 0.734 | 0.748 | 1.099 | 1.048 | 0.310 | 0.310 | 0.557 | 0.402 | 0.690 | 0.680 | 0.734 | 0.418 | 0.580 | 0.726 | 0.638 | 0.744 | 0.616 | 0.384 | 0.256 | 0.274 | 0.362 |
| | 42 | 0.362 | 0.358 | 0.360 | 0.735 | 0.754 | 1.121 | 1.058 | 0.311 | 0.311 | 0.557 | 0.401 | 0.689 | 0.679 | 0.735 | 0.419 | 0.581 | 0.724 | 0.638 | 0.745 | 0.613 | 0.387 | 0.255 | 0.276 | 0.362 |
| | 59 | 0.358 | 0.353 | 0.356 | 0.734 | 0.761 | 1.147 | 1.071 | 0.313 | 0.313 | 0.559 | 0.400 | 0.687 | 0.677 | 0.734 | 0.420 | 0.579 | 0.722 | 0.636 | 0.746 | 0.607 | 0.393 | 0.254 | 0.278 | 0.364 |
| $MRC_{nIOI}^{sw}$ | 17 | 0.380 | 0.369 | 0.375 | 0.747 | 0.733 | 1.071 | 1.035 | 0.302 | 0.302 | 0.550 | 0.412 | 0.698 | 0.684 | 0.747 | 0.419 | 0.596 | 0.723 | 0.657 | 0.772 | 0.597 | 0.403 | 0.228 | 0.277 | 0.343 |
| | 20 | 0.381 | 0.370 | 0.375 | 0.747 | 0.732 | 1.069 | 1.034 | 0.302 | 0.302 | 0.549 | 0.413 | 0.698 | 0.685 | 0.747 | 0.419 | 0.596 | 0.724 | 0.657 | 0.772 | 0.598 | 0.402 | 0.228 | 0.276 | 0.343 |
| | 23 | 0.381 | 0.370 | 0.376 | 0.747 | 0.731 | 1.070 | 1.035 | 0.302 | 0.302 | 0.549 | 0.414 | 0.698 | 0.685 | 0.747 | 0.419 | 0.596 | 0.724 | 0.657 | 0.771 | 0.599 | 0.401 | 0.229 | 0.276 | 0.343 |
| | 27 | 0.383 | 0.372 | 0.377 | 0.747 | 0.732 | 1.072 | 1.035 | 0.301 | 0.301 | 0.549 | 0.414 | 0.699 | 0.686 | 0.748 | 0.418 | 0.597 | 0.725 | 0.658 | 0.771 | 0.601 | 0.399 | 0.229 | 0.275 | 0.342 |
| | 28 | 0.383 | 0.372 | 0.377 | 0.747 | 0.732 | 1.073 | 1.036 | 0.301 | 0.301 | 0.549 | 0.414 | 0.699 | 0.686 | 0.747 | 0.418 | 0.596 | 0.725 | 0.657 | 0.770 | 0.601 | 0.399 | 0.230 | 0.275 | 0.343 |
| | 30 | 0.382 | 0.372 | 0.377 | 0.747 | 0.733 | 1.077 | 1.038 | 0.301 | 0.301 | 0.549 | 0.414 | 0.699 | 0.686 | 0.747 | 0.418 | 0.596 | 0.725 | 0.657 | 0.770 | 0.602 | 0.398 | 0.230 | 0.275 | 0.343 |
| | 37 | 0.380 | 0.370 | 0.375 | 0.746 | 0.736 | 1.085 | 1.042 | 0.302 | 0.302 | 0.550 | 0.413 | 0.698 | 0.685 | 0.746 | 0.419 | 0.595 | 0.725 | 0.656 | 0.768 | 0.602 | 0.398 | 0.232 | 0.275 | 0.344 |

*(Continued)*

Okyay and Aygun (2021), *PeerJ Comput. Sci.*, DOI 10.7717/peerj-cs.784

| Equation | BNC | Markedness | Informedness | Matthews correlation | F1-measure | MAE exact | MSE exact | RMSE exact | MAE threshold | MSE threshold | RMSE threshold | Accuracy exact | Accuracy threshold | Accuracy balanced | Fowlkes-Mallows index | Prevalence threshold | Threat score | Precision | Inverse precision | Sensitivity/Recall | Specificity | Fallout | Miss rate | False discovery rate | False omission rate |
|---|---|---|---|---|---|---|---|---|---|---|---|---|---|---|---|---|---|---|---|---|---|---|---|---|---|
| $COS_{nIOI}$ | 25 | 0.388 | 0.372 | 0.380 | 0.751 | 0.720 | 1.027 | 1.013 | 0.299 | 0.299 | 0.547 | 0.416 | 0.701 | 0.686 | 0.752 | 0.420 | 0.602 | 0.723 | 0.665 | 0.782 | 0.590 | 0.410 | 0.218 | 0.277 | 0.335 |
|  | 27 | 0.388 | 0.373 | 0.380 | 0.751 | 0.719 | 1.025 | 1.012 | 0.299 | 0.299 | 0.547 | 0.416 | 0.701 | 0.686 | 0.752 | 0.420 | 0.602 | 0.723 | 0.665 | 0.782 | 0.591 | 0.409 | 0.218 | 0.277 | 0.335 |
|  | 28 | 0.388 | 0.372 | 0.380 | 0.751 | 0.719 | 1.025 | 1.012 | 0.299 | 0.299 | 0.547 | 0.416 | 0.701 | 0.686 | 0.752 | 0.420 | 0.602 | 0.723 | 0.665 | 0.782 | 0.591 | 0.409 | 0.218 | 0.277 | 0.335 |
|  | 35 | 0.387 | 0.372 | 0.379 | 0.751 | 0.720 | 1.024 | 1.012 | 0.299 | 0.299 | 0.547 | 0.415 | 0.701 | 0.686 | 0.751 | 0.420 | 0.601 | 0.723 | 0.664 | 0.781 | 0.591 | 0.409 | 0.219 | 0.277 | 0.336 |
|  | 36 | 0.386 | 0.372 | 0.379 | 0.750 | 0.720 | 1.025 | 1.012 | 0.299 | 0.299 | 0.547 | 0.415 | 0.701 | 0.686 | 0.751 | 0.420 | 0.601 | 0.723 | 0.663 | 0.780 | 0.592 | 0.408 | 0.220 | 0.277 | 0.337 |
|  | 100 | 0.374 | 0.364 | 0.369 | 0.744 | 0.727 | 1.035 | 1.017 | 0.305 | 0.305 | 0.552 | 0.409 | 0.695 | 0.682 | 0.744 | 0.420 | 0.592 | 0.722 | 0.652 | 0.767 | 0.597 | 0.403 | 0.233 | 0.278 | 0.348 |
| $COS_{nIOI}^{sw}$ | 9 | 0.367 | 0.339 | 0.353 | 0.748 | 0.749 | 1.108 | 1.053 | 0.310 | 0.310 | 0.557 | 0.407 | 0.690 | 0.670 | 0.750 | 0.431 | 0.598 | 0.703 | 0.663 | 0.799 | 0.540 | 0.460 | 0.201 | 0.297 | 0.337 |
|  | 14 | 0.378 | 0.353 | 0.365 | 0.752 | 0.736 | 1.076 | 1.037 | 0.305 | 0.305 | 0.552 | 0.412 | 0.695 | 0.676 | 0.753 | 0.428 | 0.602 | 0.710 | 0.668 | 0.799 | 0.554 | 0.446 | 0.201 | 0.290 | 0.332 |
|  | 18 | 0.379 | 0.355 | 0.367 | 0.752 | 0.733 | 1.066 | 1.032 | 0.304 | 0.304 | 0.551 | 0.413 | 0.696 | 0.677 | 0.753 | 0.427 | 0.602 | 0.711 | 0.668 | 0.797 | 0.558 | 0.442 | 0.203 | 0.289 | 0.332 |
|  | 24 | 0.381 | 0.359 | 0.370 | 0.752 | 0.729 | 1.056 | 1.028 | 0.303 | 0.303 | 0.550 | 0.414 | 0.697 | 0.679 | 0.753 | 0.425 | 0.602 | 0.714 | 0.668 | 0.794 | 0.565 | 0.435 | 0.206 | 0.286 | 0.332 |
|  | 31 | 0.381 | 0.361 | 0.371 | 0.751 | 0.728 | 1.052 | 1.026 | 0.303 | 0.303 | 0.550 | 0.414 | 0.697 | 0.680 | 0.752 | 0.425 | 0.602 | 0.715 | 0.666 | 0.791 | 0.569 | 0.431 | 0.209 | 0.285 | 0.334 |
|  | 50 | 0.381 | 0.363 | 0.371 | 0.750 | 0.726 | 1.045 | 1.022 | 0.303 | 0.303 | 0.550 | 0.413 | 0.697 | 0.681 | 0.751 | 0.423 | 0.600 | 0.717 | 0.663 | 0.785 | 0.577 | 0.423 | 0.215 | 0.283 | 0.337 |
|  | 67 | 0.377 | 0.361 | 0.369 | 0.748 | 0.727 | 1.043 | 1.021 | 0.304 | 0.304 | 0.551 | 0.412 | 0.696 | 0.681 | 0.748 | 0.423 | 0.597 | 0.718 | 0.659 | 0.780 | 0.581 | 0.419 | 0.220 | 0.282 | 0.341 |
|  | 99 | 0.375 | 0.361 | 0.368 | 0.746 | 0.728 | 1.044 | 1.022 | 0.305 | 0.305 | 0.552 | 0.410 | 0.695 | 0.681 | 0.746 | 0.422 | 0.595 | 0.719 | 0.656 | 0.775 | 0.586 | 0.414 | 0.225 | 0.281 | 0.344 |
|  | 100 | 0.375 | 0.361 | 0.368 | 0.746 | 0.728 | 1.044 | 1.022 | 0.305 | 0.305 | 0.552 | 0.410 | 0.695 | 0.681 | 0.746 | 0.422 | 0.595 | 0.719 | 0.656 | 0.775 | 0.586 | 0.414 | 0.225 | 0.281 | 0.344 |
| $JAC_{nIOI}$ | 20 | 0.377 | 0.368 | 0.373 | 0.744 | 0.727 | 1.034 | 1.017 | 0.303 | 0.303 | 0.551 | 0.409 | 0.697 | 0.684 | 0.745 | 0.419 | 0.593 | 0.724 | 0.653 | 0.766 | 0.602 | 0.398 | 0.234 | 0.276 | 0.347 |
|  | 35 | 0.379 | 0.369 | 0.374 | 0.746 | 0.724 | 1.026 | 1.013 | 0.303 | 0.303 | 0.550 | 0.410 | 0.697 | 0.684 | 0.746 | 0.419 | 0.594 | 0.724 | 0.655 | 0.768 | 0.601 | 0.399 | 0.232 | 0.276 | 0.345 |
|  | 36 | 0.380 | 0.369 | 0.374 | 0.746 | 0.724 | 1.026 | 1.013 | 0.302 | 0.302 | 0.550 | 0.411 | 0.698 | 0.685 | 0.746 | 0.419 | 0.594 | 0.724 | 0.655 | 0.768 | 0.601 | 0.399 | 0.232 | 0.276 | 0.345 |
|  | 39 | 0.380 | 0.370 | 0.375 | 0.746 | 0.723 | 1.025 | 1.012 | 0.302 | 0.302 | 0.550 | 0.410 | 0.698 | 0.685 | 0.746 | 0.419 | 0.595 | 0.725 | 0.655 | 0.768 | 0.602 | 0.398 | 0.232 | 0.275 | 0.345 |
|  | 40 | 0.380 | 0.370 | 0.375 | 0.746 | 0.723 | 1.025 | 1.012 | 0.302 | 0.302 | 0.550 | 0.410 | 0.698 | 0.685 | 0.746 | 0.419 | 0.595 | 0.725 | 0.655 | 0.768 | 0.601 | 0.399 | 0.232 | 0.275 | 0.345 |
| $JAC_{nIOI}^{sw}$ | 18 | 0.377 | 0.360 | 0.368 | 0.748 | 0.729 | 1.050 | 1.025 | 0.304 | 0.304 | 0.552 | 0.412 | 0.696 | 0.680 | 0.749 | 0.424 | 0.597 | 0.717 | 0.660 | 0.782 | 0.578 | 0.422 | 0.218 | 0.283 | 0.340 |
|  | 29 | 0.379 | 0.363 | 0.371 | 0.748 | 0.726 | 1.042 | 1.021 | 0.303 | 0.303 | 0.551 | 0.412 | 0.697 | 0.681 | 0.749 | 0.422 | 0.598 | 0.718 | 0.661 | 0.781 | 0.582 | 0.418 | 0.219 | 0.282 | 0.339 |
|  | 32 | 0.379 | 0.364 | 0.371 | 0.748 | 0.726 | 1.041 | 1.020 | 0.303 | 0.303 | 0.550 | 0.413 | 0.697 | 0.682 | 0.749 | 0.422 | 0.598 | 0.719 | 0.661 | 0.781 | 0.583 | 0.417 | 0.219 | 0.281 | 0.339 |
|  | 36 | 0.379 | 0.364 | 0.372 | 0.748 | 0.725 | 1.039 | 1.020 | 0.303 | 0.303 | 0.550 | 0.413 | 0.697 | 0.682 | 0.749 | 0.422 | 0.598 | 0.719 | 0.660 | 0.780 | 0.584 | 0.416 | 0.220 | 0.281 | 0.340 |
|  | 44 | 0.379 | 0.364 | 0.372 | 0.748 | 0.725 | 1.038 | 1.019 | 0.303 | 0.303 | 0.550 | 0.413 | 0.697 | 0.682 | 0.748 | 0.422 | 0.597 | 0.720 | 0.660 | 0.779 | 0.586 | 0.414 | 0.221 | 0.280 | 0.340 |
|  | 45 | 0.379 | 0.364 | 0.372 | 0.748 | 0.725 | 1.038 | 1.019 | 0.303 | 0.303 | 0.550 | 0.413 | 0.697 | 0.682 | 0.748 | 0.422 | 0.597 | 0.720 | 0.659 | 0.778 | 0.586 | 0.414 | 0.222 | 0.280 | 0.341 |
|  | 59 | 0.377 | 0.363 | 0.370 | 0.747 | 0.726 | 1.038 | 1.019 | 0.304 | 0.304 | 0.551 | 0.412 | 0.696 | 0.682 | 0.747 | 0.422 | 0.596 | 0.720 | 0.657 | 0.775 | 0.588 | 0.412 | 0.225 | 0.280 | 0.343 |
|  | 100 | 0.373 | 0.361 | 0.367 | 0.744 | 0.729 | 1.042 | 1.021 | 0.306 | 0.306 | 0.553 | 0.409 | 0.694 | 0.681 | 0.745 | 0.422 | 0.593 | 0.720 | 0.653 | 0.770 | 0.591 | 0.409 | 0.230 | 0.280 | 0.347 |
| *min* |  | 0.353 | 0.339 | 0.352 | 0.728 | 0.719 | 1.024 | 1.012 | 0.296 | 0.296 | 0.544 | 0.397 | 0.684 | 0.670 | 0.728 | 0.415 | 0.572 | 0.703 | 0.628 | 0.732 | 0.540 | 0.374 | 0.201 | 0.270 | 0.332 |
| *max* |  | 0.393 | 0.379 | 0.386 | 0.753 | 0.761 | 1.147 | 1.071 | 0.316 | 0.316 | 0.562 | 0.420 | 0.704 | 0.690 | 0.753 | 0.431 | 0.604 | 0.730 | 0.668 | 0.799 | 0.626 | 0.460 | 0.268 | 0.297 | 0.372 |

**Note:**
The fractional values in the table are displayed based on three significant digits. The heat-map coloring is achieved according to full precision.

Okyay and Aygun (2021), *PeerJ Comput. Sci.*, DOI 10.7717/peerj-cs.784

**Table 11 Adequate weight-metric combination of the top-performing BNCs: ML1M.**

| Equation | BNC | Markedness | Informedness | Matthews correlation | F1-measure | MAE exact | MSE exact | RMSE exact | MAE threshold | MSE threshold | RMSE threshold | Accuracy exact | Accuracy threshold | Accuracy balanced | Fowlkes-Mallows index | Prevalence threshold | Threat score | Precision | Inverse precision | Sensitivity/Recall | Specificity | Fallout | Miss rate | False discovery rate | False omission rate |
|---|---|---|---|---|---|---|---|---|---|---|---|---|---|---|---|---|---|---|---|---|---|---|---|---|---|
| $PCC_{nIOI}$ | 4 | 0.305 | 0.308 | 0.307 | 0.735 | 0.797 | 1.200 | 1.096 | 0.328 | 0.328 | 0.572 | 0.378 | 0.672 | 0.654 | 0.735 | 0.432 | 0.581 | 0.740 | 0.566 | 0.730 | 0.578 | 0.422 | 0.270 | 0.260 | 0.434 |
| | 31 | 0.396 | 0.376 | 0.386 | 0.779 | 0.707 | 0.981 | 0.991 | 0.283 | 0.283 | 0.532 | 0.415 | 0.717 | 0.688 | 0.780 | 0.422 | 0.639 | 0.755 | 0.641 | 0.805 | 0.571 | 0.429 | 0.195 | 0.245 | 0.359 |
| | 91 | 0.413 | 0.382 | 0.397 | 0.787 | 0.696 | 0.966 | 0.983 | 0.277 | 0.277 | 0.526 | 0.422 | 0.723 | 0.691 | 0.788 | 0.423 | 0.649 | 0.754 | 0.659 | 0.824 | 0.559 | 0.441 | 0.176 | 0.246 | 0.341 |
| | 96 | 0.412 | 0.382 | 0.397 | 0.787 | 0.696 | 0.966 | 0.983 | 0.277 | 0.277 | 0.526 | 0.423 | 0.723 | 0.691 | 0.788 | 0.423 | 0.649 | 0.754 | 0.659 | 0.824 | 0.557 | 0.443 | 0.176 | 0.246 | 0.341 |
| | 99 | 0.413 | 0.382 | 0.397 | 0.787 | 0.696 | 0.966 | 0.983 | 0.277 | 0.277 | 0.526 | 0.422 | 0.723 | 0.691 | 0.788 | 0.423 | 0.649 | 0.754 | 0.659 | 0.825 | 0.557 | 0.443 | 0.175 | 0.246 | 0.341 |
| | 100 | 0.413 | 0.382 | 0.397 | 0.788 | 0.696 | 0.966 | 0.983 | 0.277 | 0.277 | 0.526 | 0.423 | 0.723 | 0.691 | 0.788 | 0.423 | 0.650 | 0.754 | 0.659 | 0.825 | 0.557 | 0.443 | 0.175 | 0.246 | 0.341 |
| $PCC_{nIOI}^{sw}$ | 31 | 0.444 | 0.368 | 0.404 | 0.801 | 0.688 | 0.988 | 0.994 | 0.270 | 0.270 | 0.520 | 0.442 | 0.730 | 0.684 | 0.804 | 0.432 | 0.668 | 0.740 | 0.704 | 0.873 | 0.495 | 0.505 | 0.127 | 0.260 | 0.296 |
| | 93 | 0.439 | 0.370 | 0.403 | 0.799 | 0.687 | 0.981 | 0.990 | 0.271 | 0.271 | 0.520 | 0.440 | 0.729 | 0.685 | 0.802 | 0.431 | 0.666 | 0.741 | 0.698 | 0.868 | 0.502 | 0.498 | 0.132 | 0.259 | 0.302 |
| | 100 | 0.439 | 0.370 | 0.403 | 0.799 | 0.687 | 0.979 | 0.990 | 0.271 | 0.271 | 0.520 | 0.440 | 0.729 | 0.685 | 0.802 | 0.431 | 0.666 | 0.741 | 0.698 | 0.867 | 0.502 | 0.498 | 0.133 | 0.259 | 0.302 |
| $MRC_{nIOI}$ | 8 | 0.351 | 0.344 | 0.347 | 0.759 | 0.747 | 1.075 | 1.037 | 0.304 | 0.304 | 0.552 | 0.398 | 0.696 | 0.672 | 0.759 | 0.426 | 0.611 | 0.748 | 0.603 | 0.769 | 0.575 | 0.425 | 0.231 | 0.252 | 0.397 |
| | 48 | 0.403 | 0.379 | 0.391 | 0.783 | 0.704 | 0.980 | 0.990 | 0.280 | 0.280 | 0.530 | 0.417 | 0.720 | 0.689 | 0.784 | 0.422 | 0.643 | 0.754 | 0.649 | 0.814 | 0.564 | 0.436 | 0.186 | 0.246 | 0.351 |
| | 60 | 0.406 | 0.379 | 0.393 | 0.785 | 0.702 | 0.978 | 0.989 | 0.279 | 0.279 | 0.528 | 0.419 | 0.721 | 0.690 | 0.785 | 0.423 | 0.646 | 0.754 | 0.652 | 0.818 | 0.562 | 0.438 | 0.182 | 0.246 | 0.348 |
| | 92 | 0.411 | 0.380 | 0.395 | 0.787 | 0.701 | 0.980 | 0.990 | 0.278 | 0.278 | 0.527 | 0.421 | 0.722 | 0.690 | 0.787 | 0.423 | 0.648 | 0.753 | 0.657 | 0.823 | 0.557 | 0.443 | 0.177 | 0.247 | 0.343 |
| | 97 | 0.411 | 0.380 | 0.395 | 0.787 | 0.701 | 0.979 | 0.990 | 0.278 | 0.278 | 0.527 | 0.420 | 0.722 | 0.690 | 0.787 | 0.423 | 0.648 | 0.753 | 0.657 | 0.823 | 0.557 | 0.443 | 0.177 | 0.247 | 0.343 |
| | 100 | 0.410 | 0.380 | 0.395 | 0.787 | 0.701 | 0.980 | 0.990 | 0.278 | 0.278 | 0.527 | 0.420 | 0.722 | 0.690 | 0.787 | 0.423 | 0.648 | 0.753 | 0.657 | 0.823 | 0.556 | 0.444 | 0.177 | 0.247 | 0.343 |
| $MRC_{nIOI}^{sw}$ | 23 | 0.434 | 0.364 | 0.397 | 0.798 | 0.698 | 1.001 | 1.001 | 0.273 | 0.273 | 0.523 | 0.434 | 0.727 | 0.682 | 0.800 | 0.432 | 0.664 | 0.739 | 0.695 | 0.867 | 0.497 | 0.503 | 0.133 | 0.261 | 0.305 |
| | 41 | 0.434 | 0.366 | 0.399 | 0.798 | 0.694 | 0.992 | 0.996 | 0.273 | 0.273 | 0.522 | 0.435 | 0.727 | 0.683 | 0.800 | 0.432 | 0.664 | 0.740 | 0.693 | 0.865 | 0.501 | 0.499 | 0.135 | 0.260 | 0.307 |
| | 44 | 0.434 | 0.367 | 0.400 | 0.798 | 0.694 | 0.992 | 0.996 | 0.272 | 0.272 | 0.522 | 0.435 | 0.728 | 0.684 | 0.800 | 0.431 | 0.664 | 0.741 | 0.694 | 0.865 | 0.502 | 0.498 | 0.135 | 0.259 | 0.306 |
| | 58 | 0.433 | 0.368 | 0.399 | 0.798 | 0.694 | 0.990 | 0.995 | 0.272 | 0.272 | 0.522 | 0.435 | 0.728 | 0.684 | 0.800 | 0.431 | 0.663 | 0.741 | 0.692 | 0.863 | 0.505 | 0.495 | 0.137 | 0.259 | 0.308 |
| | 93 | 0.431 | 0.368 | 0.398 | 0.797 | 0.694 | 0.989 | 0.995 | 0.273 | 0.273 | 0.522 | 0.434 | 0.727 | 0.684 | 0.799 | 0.431 | 0.662 | 0.742 | 0.689 | 0.861 | 0.508 | 0.492 | 0.139 | 0.258 | 0.311 |
| | 94 | 0.431 | 0.368 | 0.398 | 0.797 | 0.694 | 0.989 | 0.995 | 0.273 | 0.273 | 0.522 | 0.434 | 0.727 | 0.684 | 0.799 | 0.431 | 0.662 | 0.742 | 0.689 | 0.861 | 0.508 | 0.492 | 0.139 | 0.258 | 0.311 |
| | 96 | 0.431 | 0.368 | 0.398 | 0.797 | 0.694 | 0.990 | 0.995 | 0.273 | 0.273 | 0.522 | 0.434 | 0.727 | 0.684 | 0.799 | 0.431 | 0.662 | 0.742 | 0.689 | 0.860 | 0.508 | 0.492 | 0.140 | 0.258 | 0.311 |
| $COS_{nIOI}$ | 4 | 0.373 | 0.346 | 0.359 | 0.774 | 0.736 | 1.070 | 1.034 | 0.294 | 0.294 | 0.542 | 0.410 | 0.706 | 0.673 | 0.775 | 0.431 | 0.631 | 0.742 | 0.631 | 0.809 | 0.537 | 0.463 | 0.191 | 0.258 | 0.369 |
| | 13 | 0.418 | 0.372 | 0.394 | 0.792 | 0.692 | 0.969 | 0.984 | 0.276 | 0.276 | 0.525 | 0.430 | 0.724 | 0.686 | 0.793 | 0.428 | 0.655 | 0.746 | 0.673 | 0.843 | 0.528 | 0.472 | 0.157 | 0.254 | 0.327 |
| | 36 | 0.430 | 0.375 | 0.402 | 0.796 | 0.684 | 0.951 | 0.975 | 0.272 | 0.272 | 0.522 | 0.434 | 0.728 | 0.687 | 0.798 | 0.428 | 0.661 | 0.745 | 0.685 | 0.854 | 0.520 | 0.480 | 0.146 | 0.255 | 0.315 |
| | 41 | 0.431 | 0.374 | 0.402 | 0.796 | 0.683 | 0.950 | 0.975 | 0.272 | 0.272 | 0.522 | 0.434 | 0.728 | 0.687 | 0.798 | 0.428 | 0.661 | 0.745 | 0.686 | 0.855 | 0.520 | 0.480 | 0.145 | 0.255 | 0.314 |
| | 45 | 0.430 | 0.374 | 0.401 | 0.796 | 0.683 | 0.949 | 0.974 | 0.272 | 0.272 | 0.522 | 0.434 | 0.728 | 0.687 | 0.798 | 0.429 | 0.661 | 0.745 | 0.685 | 0.855 | 0.519 | 0.481 | 0.145 | 0.255 | 0.315 |
| | 50 | 0.430 | 0.374 | 0.401 | 0.796 | 0.683 | 0.951 | 0.975 | 0.272 | 0.272 | 0.522 | 0.434 | 0.728 | 0.687 | 0.798 | 0.429 | 0.661 | 0.745 | 0.686 | 0.855 | 0.518 | 0.482 | 0.145 | 0.255 | 0.314 |
| | 72 | 0.429 | 0.372 | 0.399 | 0.796 | 0.684 | 0.954 | 0.977 | 0.273 | 0.273 | 0.522 | 0.435 | 0.727 | 0.686 | 0.798 | 0.429 | 0.661 | 0.744 | 0.685 | 0.855 | 0.517 | 0.483 | 0.145 | 0.256 | 0.315 |
| | 83 | 0.429 | 0.371 | 0.399 | 0.796 | 0.684 | 0.954 | 0.977 | 0.273 | 0.273 | 0.523 | 0.434 | 0.727 | 0.686 | 0.798 | 0.429 | 0.661 | 0.744 | 0.685 | 0.855 | 0.516 | 0.484 | 0.145 | 0.256 | 0.315 |

(Continued)

Okyay and Aygun (2021), *PeerJ Comput. Sci.*, DOI 10.7717/peerj-cs.784

| Equation | BNC | Markedness | Informedness | Matthews correlation | F1-measure | MAE exact | MSE exact | RMSE exact | MAE threshold | MSE threshold | RMSE threshold | Accuracy exact | Accuracy threshold | Accuracy balanced | Fowlkes-Mallows index | Prevalence threshold | Threat score | Precision | Inverse precision | Sensitivity/Recall | Specificity | Fallout | Miss rate | False discovery rate | False omission rate |
|---|---|---|---|---|---|---|---|---|---|---|---|---|---|---|---|---|---|---|---|---|---|---|---|---|---|
| $COS_{nIOI}^{sw}$ | 18 | 0.428 | 0.343 | 0.383 | 0.797 | 0.712 | 1.044 | 1.022 | 0.278 | 0.278 | 0.527 | 0.430 | 0.722 | 0.672 | 0.800 | 0.438 | 0.662 | 0.730 | 0.699 | 0.878 | 0.466 | 0.534 | 0.122 | 0.270 | 0.301 |
| | 30 | 0.430 | 0.347 | 0.387 | 0.797 | 0.706 | 1.030 | 1.015 | 0.277 | 0.277 | 0.526 | 0.433 | 0.723 | 0.674 | 0.801 | 0.437 | 0.663 | 0.731 | 0.699 | 0.877 | 0.470 | 0.530 | 0.123 | 0.269 | 0.301 |
| | 72 | 0.429 | 0.352 | 0.389 | 0.797 | 0.701 | 1.015 | 1.007 | 0.276 | 0.276 | 0.526 | 0.434 | 0.724 | 0.676 | 0.800 | 0.436 | 0.662 | 0.733 | 0.696 | 0.873 | 0.479 | 0.521 | 0.127 | 0.267 | 0.304 |
| | 91 | 0.428 | 0.352 | 0.388 | 0.796 | 0.701 | 1.012 | 1.006 | 0.277 | 0.277 | 0.526 | 0.434 | 0.723 | 0.676 | 0.799 | 0.436 | 0.662 | 0.734 | 0.694 | 0.871 | 0.481 | 0.519 | 0.129 | 0.266 | 0.306 |
| | 92 | 0.428 | 0.352 | 0.388 | 0.796 | 0.701 | 1.012 | 1.006 | 0.277 | 0.277 | 0.526 | 0.434 | 0.723 | 0.676 | 0.799 | 0.436 | 0.662 | 0.734 | 0.694 | 0.871 | 0.482 | 0.518 | 0.129 | 0.266 | 0.306 |
| | 97 | 0.427 | 0.352 | 0.388 | 0.796 | 0.701 | 1.011 | 1.006 | 0.277 | 0.277 | 0.526 | 0.434 | 0.723 | 0.676 | 0.799 | 0.436 | 0.661 | 0.734 | 0.693 | 0.870 | 0.482 | 0.518 | 0.130 | 0.266 | 0.307 |
| | 99 | 0.427 | 0.352 | 0.388 | 0.796 | 0.701 | 1.011 | 1.005 | 0.277 | 0.277 | 0.526 | 0.433 | 0.723 | 0.676 | 0.799 | 0.435 | 0.661 | 0.734 | 0.693 | 0.870 | 0.482 | 0.518 | 0.130 | 0.266 | 0.307 |
| | 100 | 0.427 | 0.352 | 0.388 | 0.796 | 0.701 | 1.011 | 1.005 | 0.277 | 0.277 | 0.526 | 0.433 | 0.723 | 0.676 | 0.799 | 0.436 | 0.661 | 0.734 | 0.693 | 0.870 | 0.482 | 0.518 | 0.130 | 0.266 | 0.307 |
| $JAC_{nIOI}$ | 10 | 0.396 | 0.369 | 0.382 | 0.781 | 0.705 | 0.983 | 0.991 | 0.284 | 0.284 | 0.533 | 0.419 | 0.716 | 0.685 | 0.782 | 0.425 | 0.641 | 0.750 | 0.646 | 0.814 | 0.555 | 0.445 | 0.186 | 0.250 | 0.354 |
| | 29 | 0.417 | 0.382 | 0.399 | 0.790 | 0.689 | 0.947 | 0.973 | 0.275 | 0.275 | 0.524 | 0.425 | 0.725 | 0.691 | 0.791 | 0.424 | 0.652 | 0.752 | 0.665 | 0.831 | 0.551 | 0.449 | 0.169 | 0.248 | 0.335 |
| | 53 | 0.420 | 0.382 | 0.400 | 0.791 | 0.687 | 0.942 | 0.971 | 0.274 | 0.274 | 0.524 | 0.426 | 0.726 | 0.691 | 0.792 | 0.424 | 0.654 | 0.752 | 0.668 | 0.834 | 0.548 | 0.452 | 0.166 | 0.248 | 0.332 |
| | 58 | 0.420 | 0.382 | 0.400 | 0.791 | 0.687 | 0.943 | 0.971 | 0.274 | 0.274 | 0.524 | 0.426 | 0.726 | 0.691 | 0.792 | 0.424 | 0.654 | 0.752 | 0.668 | 0.834 | 0.547 | 0.453 | 0.166 | 0.248 | 0.332 |
| | 84 | 0.420 | 0.381 | 0.400 | 0.791 | 0.688 | 0.944 | 0.972 | 0.274 | 0.274 | 0.524 | 0.426 | 0.726 | 0.690 | 0.792 | 0.424 | 0.654 | 0.751 | 0.668 | 0.835 | 0.546 | 0.454 | 0.165 | 0.249 | 0.332 |
| | 90 | 0.419 | 0.380 | 0.399 | 0.791 | 0.688 | 0.945 | 0.972 | 0.275 | 0.275 | 0.524 | 0.426 | 0.725 | 0.690 | 0.792 | 0.425 | 0.654 | 0.751 | 0.668 | 0.835 | 0.545 | 0.455 | 0.165 | 0.249 | 0.332 |
| | 99 | 0.419 | 0.380 | 0.399 | 0.791 | 0.688 | 0.945 | 0.972 | 0.275 | 0.275 | 0.524 | 0.426 | 0.725 | 0.690 | 0.792 | 0.425 | 0.654 | 0.751 | 0.668 | 0.836 | 0.544 | 0.456 | 0.164 | 0.249 | 0.332 |
| $JAC_{nIOI}^{sw}$ | 4 | 0.366 | 0.333 | 0.349 | 0.773 | 0.745 | 1.098 | 1.048 | 0.297 | 0.297 | 0.545 | 0.407 | 0.703 | 0.667 | 0.774 | 0.435 | 0.630 | 0.735 | 0.631 | 0.815 | 0.518 | 0.482 | 0.185 | 0.265 | 0.369 |
| | 25 | 0.422 | 0.366 | 0.393 | 0.794 | 0.692 | 0.972 | 0.986 | 0.276 | 0.276 | 0.525 | 0.431 | 0.724 | 0.683 | 0.795 | 0.431 | 0.658 | 0.742 | 0.680 | 0.853 | 0.513 | 0.487 | 0.147 | 0.258 | 0.320 |
| | 54 | 0.425 | 0.366 | 0.395 | 0.795 | 0.688 | 0.965 | 0.982 | 0.275 | 0.275 | 0.524 | 0.433 | 0.725 | 0.683 | 0.797 | 0.431 | 0.659 | 0.742 | 0.683 | 0.856 | 0.510 | 0.490 | 0.144 | 0.258 | 0.317 |
| | 55 | 0.425 | 0.366 | 0.394 | 0.795 | 0.688 | 0.965 | 0.982 | 0.275 | 0.275 | 0.524 | 0.433 | 0.725 | 0.683 | 0.797 | 0.431 | 0.659 | 0.742 | 0.683 | 0.856 | 0.510 | 0.490 | 0.144 | 0.258 | 0.317 |
| | 56 | 0.425 | 0.366 | 0.395 | 0.795 | 0.688 | 0.965 | 0.982 | 0.275 | 0.275 | 0.524 | 0.433 | 0.725 | 0.683 | 0.797 | 0.431 | 0.659 | 0.742 | 0.683 | 0.856 | 0.510 | 0.490 | 0.144 | 0.258 | 0.317 |
| | 75 | 0.425 | 0.366 | 0.394 | 0.795 | 0.689 | 0.966 | 0.983 | 0.275 | 0.275 | 0.524 | 0.433 | 0.725 | 0.683 | 0.797 | 0.431 | 0.659 | 0.741 | 0.683 | 0.856 | 0.509 | 0.491 | 0.144 | 0.259 | 0.317 |
| *min* | | 0.305 | 0.308 | 0.307 | 0.735 | 0.683 | 0.942 | 0.971 | 0.270 | 0.270 | 0.520 | 0.378 | 0.672 | 0.654 | 0.735 | 0.422 | 0.581 | 0.730 | 0.566 | 0.730 | 0.466 | 0.422 | 0.122 | 0.245 | 0.296 |
| *max* | | 0.444 | 0.382 | 0.404 | 0.801 | 0.797 | 1.200 | 1.096 | 0.328 | 0.328 | 0.572 | 0.442 | 0.730 | 0.691 | 0.804 | 0.438 | 0.668 | 0.755 | 0.704 | 0.878 | 0.578 | 0.534 | 0.270 | 0.270 | 0.434 |

**Note:**
The fractional values in the table are displayed based on three significant digits. The heat-map coloring is achieved according to full precision.

*BNC* range. The homogeneous column-wise scoring highlights the *BNC*-free performance of any similarity equation. For instance, the *JAC* equation with *SW* generally maintained its stability in each metric group, considering smooth coloring. Remarkably, homogeneous scoring highlights the overall performance of any similarity equation.

In a general view, if an RS design targets only the recommendations of preferable items, the *COS* may be a suitable similarity measure. Metrics that do not address *TN* values such as *F1-measure*, *Fowlkes–Mallows index*, *threat score*, *sensitivity*, and *miss rate* feedback the indicative scores when combined with the *COS*. Conversely, the *COS* with the *SW* approach, exhibited a homogeneity of the heat-map tones, which deteriorates while transitioning between two correlation methods. Although some metrics have the advantage of *SW*, the *COS* is not completely compatible with the *SW* approach. Conversely, the beneficial impact of the *SW* can be observed through all the other equations. Overall, the *PCC* is the most appealing metric. Although the *PCC* considers linear correlation, it suitably fits into the five-star rating analyses. *PCC* with *SW* can be generalized as one of the adequate similarity equations, whereas *MRC* without *SW* became the least performing equation as the harsh red background color indicates. The *MRC* without *SW* showed the most inadequate performance as shown in Table 10; therefore, the weighting method for the *MRC* utilization is highly recommended for the ML100K, whereas the case is slightly different for the ML1M.

## CONCLUSIONS

This paper presented an experimental perspective for interpreting the interrelations between similarity equations and performance metrics. The most indicative highlight of this article is the necessity of a dynamic approach by performing independent computations. The misleading effect of test-item bias has been emphasized in our analyses. It has been unveiled how this pitfall can demarcate the results. The test-item bias in the training phase results in hazardous outcomes, and the upper limit a system can reach was determined. Another highlight is the impact of similarity weighting. All combinations of the modifications were monitored experimentally. The overall evaluation was inferred from multiple simulations. In addition, we have conducted a fine-tuned neighborhood analysis on the weight-metric combinations. The limit of the *BNC* can be deduced from our graphical interpretations. Furthermore, our remarks have been profoundly demonstrated in the heat-map tables with the best-performing neighborhood surveyed by using intercrossing similarity equations and specific metrics. Overall, the back-end of any RS design can be developed using the same procedures applied throughout this study. Any dataset can be adaptively examined *via* the open-source code (as indicated in the Acknowledgments section) of our framework. Starting from the fine-tuned neighborhood, the test-item bias mitigation approach can be thoroughly followed *with* and *without* the *SW* method. We believe that further studies in RS science can benefit from the findings of this study. Considering future studies, any other metadata or features, such as user demographics and item details, can be included to enhance our framework.

## ACKNOWLEDGEMENTS

A reproducible code run can be directly executed on the Code Ocean platform. Moreover, both the open-source code of the whole package and the supplementary material, supplementaryMaterial_allResults.xlsx, have been attached to the https://github.com/savasokyay/AdequateWeightMetricDynamicCF repository. The supplemental material contains detailed results, including all the neighbors from the top performing to the least performing metrics. This file can easily be adapted by filtering and sorting the fields.

### Funding

The authors received no funding for this work.

### Competing Interests
The authors declare that they have no competing interests.

### Author Contributions
- Savas Okyay conceived and designed the experiments, performed the experiments, analyzed the data, performed the computation work, prepared figures and/or tables, and approved the final draft.
- Sercan Aygun conceived and designed the experiments, performed the experiments, analyzed the data, authored or reviewed drafts of the paper, and approved the final draft.

### Data Availability
A reproducible code run can be directly executed on the Code Ocean platform. Both the open-source code of the whole package and result file are available at GitHub: https://github.com/savasokyay/AdequateWeightMetricDynamicCF.

The data used throughout the experiments are available at ML100K release: https://grouplens.org/datasets/movielens/100k/ and at ML1M release: https://grouplens.org/datasets/movielens/1m/.

### Supplemental Information
Supplemental information for this article can be found online at http://dx.doi.org/10.7717/peerj-cs.784#supplemental-information.

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
