# Peer review of "Experimental interpretation of adequate weight-metric combination for dynamic user-based collaborative filtering"

_PeerJ Computer Science, doi:10.7717/peerj-cs.784_

## Round 0.1 · original submission · Major Revisions

Both reviewers raised major concerns. As a result, an essential revision is recommended.

Reviewer 1 ·

Basic reporting

The manuscript contains a lot of grammatical inadequacies which makes it practically impossible for me to logically follow its flow.

I will recommend that the authors should seek the assistance of a native English speaker to help them in editing the manuscript as it cannot be publish in its present form.

The entire manuscript should be written in good English language and re-submitted for review.

Experimental design

Manuscript has to be re-written first in good English language before comments can be made in this section.

Validity of the findings

Manuscript has to be re-written first in good English before comments can be made in this section.

Additional comments

Manuscript has to be re-written first in good English language before comments can be made in this section.

·

Basic reporting

The language needs to be improved. A non-exhaustive list of stylistic glitches and suggestions are:
-- Abstract
- 24 "recommendations that.." --> "finding recommendations which appeal to each user varies"
- 26 "it is measured" --> "we measure the appropriateness of the recommendation in terms of"
-- Introduction
- 58 "there are loads of" --> "there are many reported RS implementations"
- 58 "is blurry" --> "it is unclear, how the "
--- Getting started to experiments --> "Experimental design"

Other issues similar to the above.

The figures Fig 1 to 5 are illegible and it is impossible to verify the conclusions drawn from them.

Experimental design

The authors have championed the cause of leave one out methods, however, there should be a discussion of standard statistical augmented methods like ANOVA; that is, the approach discussed here is only true under the severe assumption that each covariate is independent of the others.

It is unclear how the median is considered to reduce the outliers; standard techniques to identify (like checking residuals and standardized residuals) should be compared. The median values are indeed better than the mean under certain scenarios, however, it is best to recall, that there are no general unbiased estimators for determining the population median.

Note that the Pearson correlation is based on the mean and so the estimator is actually an unbiased estimate of the population statistics.

Validity of the findings

It is unclear why the smooth coloring of the JAC with SW is of merit. In general, the distribution of metrics is unclear as a statistic of interest, as it is sensitive to the order of the table and the dataset. If the question is of tracking the metrics themselves, then it would be better to describe the data in terms of a density plot.

The data and code is provided, and this is commendable.

Additional comments

The research question posed is appropriate and the code is sufficient for the analysis of the same. However, the manuscript at this stage is not ready for publication and needs to be reworked for clarity. Several key points raised in section 2 in particular need to be addressed adequately. The paper contains implementation details and shows significant effort in terms of covering the existing metrics. I am certain it will be a suitable addition to the literature after revisions.

---

## Round 0.2 · Major Revisions

The concerns highlighted must be addressed. Please see the reviewers' comments.

·

Basic reporting

The paper's readability has improved significantly. However, I did not intend to imply that the paper needed an editorial service; and I am sorry the authors had to retain one.

Experimental design

N/A.

Validity of the findings

N/A.

Additional comments

The authors have addressed my previous comments; and the paper has been extensively reworked. At this point I am amenable towards this paper's acceptance.

Reviewer 3 ·

Basic reporting

The structure of manuscript is not well-organized, and the content is not coherent, making it less readable.

Experimental design

The procedure of the experiment and the settings of the parameters are described with sufficient detail. Meanwhile, the meaning of each metric is illustrated in detail.

Validity of the findings

The dpi of the figures in the manuscript is too low. After downloading the figures from the attachment, some results can be verified. However, the reviewer cannot correspond to the ”BNC=10” in line 532, where did the authors get it? It should be emphasized that the discussion is difficult to be well understood, and it is necessary to report the results in a more structured manner.

Additional comments

1) The merits and demerits of the four touchstone similarity equations should be introduced in detail.
2) There are 8 formulas in the section of experimental design, the authors only analysed the PCC, why not discuss the other optimal BNCs?
3) On line 202 page 10, the way of citation is inconsistent with the previous content, please double-check.
4) Some abbreviations are defined repeatedly (e.g., PCC, MRC, COS, JAC in section ITEM INDEPENDENCY ANALYSES)

---

## Author Rebuttal · Round 0.2

**Computer Engineering Dept.**
Yildiz Technical University
Electrical Electronics Fac.
D-026 Esenler, Istanbul
TURKEY, 34220
Tel: +90-555-806-97-18
https://www.yildiz.edu.tr/en
ayguns@yildiz.edu.tr

August 18ᵗʰ, 2021

Dear Editor

Thank you for giving us the opportunity to submit a revised draft of our manuscript titled "Experimental interpretation of adequate weight-metric combination for dynamic user-based collaborative filtering" to **PeerJ Computer Science**. We appreciate the time and effort that you and the reviewers have dedicated to providing your valuable feedback on the manuscript. We are grateful to the reviewers for their insightful comments on our paper. We have been able to incorporate changes to reflect most of the suggestions provided by the reviewers. Here is a point-by-point response to the reviewers' comments and concerns.

Best Regards,

Res. Asst. Sercan AYGUN
Yildiz Technical University, Computer Engineering Dept.

On behalf of all authors.

# Reviewer 1

## Basic reporting

*The manuscript contains a lot of grammatical inadequacies which makes it practically impossible for me to logically follow its flow.*

*I will recommend that the authors should seek the assistance of a native English speaker to help them in editing the manuscript as it cannot be publish in its present form.*

*The entire manuscript should be written in good English language and re-submitted for review.*

## Experimental design

*Manuscript has to be re-written first in good English language before comments can be made in this section.*

## Validity of the findings

*Manuscript has to be re-written first in good English before comments can be made in this section.*

## Additional comments

*Manuscript has to be re-written first in good English language before comments can be made in this section.*

## Author Response

Thank you for your recommendation. The paper was edited by the *Editage* language service and the paper has now arrived at the error-free version. The editing certificate can be found on the next page. Besides, several technical enhancements were accomplished related to the outlier analyses & independence checks, which were the main feedback of Reviewer-2.

We hope that the current version of our manuscript is ready for publication. Thank you very much for your time and consideration.

# Editing Certificate

This document certifies that the manuscript listed below has been edited to ensure language and grammar accuracy and is error free in these aspects. The logical presentation of ideas and the structure of the paper were also checked during the editing process. The edit was performed by professional editors at Editage, a division of Cactus Communications. The author's core research ideas were not altered in any way during the editing process. The quality of the edit has been guaranteed, with the assumption that our suggested changes have been accepted and the text has not been further altered without the knowledge of our editors.

MANUSCRIPT TITLE

**Experimental Interpretation of Adequate Weight-metric Combination for Dynamic User-based Collaborative Filtering**

AUTHORS

**Savas Okyay, Sercan Aygun**

ISSUED ON

**August 16, 2021**

JOB CODE

**SAOKY_1**

[Figure]

*Vikas Narang*

**Vikas Narang**
**Chief Operating Officer - Editage**

# editage

Editage, a brand of Cactus Communications, offers professional English language editing and publication support services to authors engaged in over 1300 areas of research. Through its community of experienced editors, which includes doctors, engineers, published scientists, and researchers with peer review experience, Editage has successfully helped authors get published in internationally reputed journals. Authors who work with Editage are guaranteed excellent language quality and timely delivery.

**GLOBAL :**
(833) 979-0061 | request@editage.com

**CHINA :**
400-120-3020 | fabiao@editage.cn

## CACTUS

IMPACT SCIENCE — impact.science

R — researcher.life

CACTUS — lifesciences.cactusglobal.com

## Reviewer 2 - Rohit Goswami

### Basic reporting #1

*The language needs to be improved. A non-exhaustive list of stylistic glitches and suggestions are:*

*-- Abstract*

*- 24 "recommendations that.." --> "finding recommendations which appeal to each user varies"*

*- 26 "it is measured" --> "we measure the appropriateness of the recommendation in terms of"*

*-- Introduction*

*- 58 "there are loads of" --> "there are many reported RS implementations"*

*- 58 "is blurry" --> "it is unclear, how the "*

*--- Getting started to experiments --> "Experimental design"*

*Other issues similar to the above.*

## Author Response

Thank you very much for your feedback. First of all, your suggestions were performed. In addition, referring to Reviewer-1's request for language editing, our article has been sent to a language editing service. We hope this now resolves all language issues you mentioned and may speed up the article publication process in case of possible acceptance. The certificate is attached to the next page.

# Editing Certificate

This document certifies that the manuscript listed below has been edited to ensure language and grammar accuracy and is error free in these aspects. The logical presentation of ideas and the structure of the paper were also checked during the editing process. The edit was performed by professional editors at Editage, a division of Cactus Communications. The author's core research ideas were not altered in any way during the editing process. The quality of the edit has been guaranteed, with the assumption that our suggested changes have been accepted and the text has not been further altered without the knowledge of our editors.

MANUSCRIPT TITLE

**Experimental Interpretation of Adequate Weight-metric Combination for Dynamic User-based Collaborative Filtering**

AUTHORS

**Savas Okyay, Sercan Aygun**

ISSUED ON

**August 16, 2021**

JOB CODE

**SAOKY_1**

[Figure]

*Vikas Narang*

**Vikas Narang**
**Chief Operating Officer - Editage**

## editage

Editage, a brand of Cactus Communications, offers professional English language editing and publication support services to authors engaged in over 1300 areas of research. Through its community of experienced editors, which includes doctors, engineers, published scientists, and researchers with peer review experience, Editage has successfully helped authors get published in internationally reputed journals. Authors who work with Editage are guaranteed excellent language quality and timely delivery.

**GLOBAL :**
(833) 979-0061  |  request@editage.com

**CHINA :**
400-120-3020  |  fabiao@editage.cn

## CACTUS

IMPACT SCIENCE   impact.science

R   researcher.life

CACTUS   lifesciences.cactusglobal.com

*Basic reporting #2*
*The figures Fig 1 to 5 are illegible and it is impossible to verify the conclusions drawn from them.*

## Author Response

All figure files seem to have a low resolution unintentionally in the auto-generated document because the PeerJ article processing system automatically reduces the figure quality. While generating the related figures, we had produced them in ultra-high quality. In fact, in the first round, a note related to this was given in the "associated data" section; as shown below, high-quality images had been added to the supplementary material section as an extra.

[Figure]

Moreover, original versions of the images and the supplementary files can be accessed from the "Primary Files" section on the submitted manuscript page with full resolution and precision. The following image is captured from the author submission panel.

[Figure]

Nevertheless, for this round, high-quality versions of the images (and also all submission files in our local repository) are included in the following cloud link for quick access from this document. Link: https://1drv.ms/u/s!AhotH2rU6kw_itEy-yLk5Nspb7d3KQ?e=jKueGm

***Experimental design #1***

*The authors have championed the cause of leave one out methods, however, there should be a discussion of standard statistical augmented methods like ANOVA; that is, the approach discussed here is only true under the severe assumption that each covariate is independent of the others.*

## Author Response

First of all, we would like to thank you very much for your comment that increases the impact of our article. We totally agree with your comment. For this reason, we completed our analysis with ANOVA, as you suggested.

As you have underlined, the condition of item independencies is critical for the validity of the proposed approach. For this purpose, each dataset with the user × item format was subjected to variance analysis to prove that each column is independent (uncorrelated) from any other. Since no user group or additional information (demographics, movie specifications like genre, etc.) was used, analyses were completed through one-way ANOVA.

ANOVA supplies information about between-groups variation (`Groups`) and within-groups variation (`Error`). As in the following *Response Table 1* (a), we present the *ANOVA Table* of each dataset. By calculating the sum of squares (SS), the degrees of freedom (df), thereby the mean squared errors (MS), the F-test is applied. The ratio of between-group (inter-) variability and within-group (intra-) variability is obtained, which is analyzed in the context of the *null hypothesis*. By showing F values greater than 1, we complete the first step of item independence validation. Moreover, the distribution of F is observed to measure the probability value to further guarantee not all the means are the same by F>>1 values and the probability (P) values, which are obtained by the F-distribution. The lower the P-values are, the higher chances of strong evidence against the *null hypothesis*. Besides, we also present visual plots of some selected items in *Response Table 1* (b). We depict the randomly selected items showing the box plot analysis related to the *median*, *minimum* and *maximum* values, *inner-quartile range* of 75th and 25th percentiles, 95% of *upper confidence limit of the median,* and 25% of *lower confidence limit of the median*. The x-axis shows the unit rating values, and the y-axis shows the randomly selected item IDs. Furthermore, we also depict the analysis of all items using the complete box plot in the following *Response Table 1* (c).

# Response Table 1: One-way ANOVA analyses of ML100K and ML1M

[Figure]

| ML100K | ML1M |
|--------|------|

**ANOVA Table** (ML100K)

| Source | SS | df | MS | F | Prob>F |
|--------|-----|-----|-----|-----|--------|
| Groups | 26698.8 | 1681 | 15.8827 | 15.61 | 0 |
| Error | 100014 | 98318 | 1.0173 | | |
| Total | 126712.8 | 99999 | | | |

**ANOVA Table** (ML1M)

| Source | SS | df | MS | F | Prob>F |
|--------|-----|-----|-----|-----|--------|
| Groups | 297914.9 | 3705 | 80.4089 | 84.32 | 0 |
| Error | 950261.2 | 996503 | 0.9536 | | |
| Total | 1248176.1 | 1000208 | | | |

*(a)* *ANOVA stats for the whole dataset*

*(b)* *Item independence examples with randomly selected items*

*(c)* *Item independences for all items*

All in all, we now proved that the validity of our approach by showing how the uncorrelated of each item. The related explanations were added to revised manuscript, *Materials & Methods Section,* shown as follows. Table 3 and Table 4 in the revised manuscript were also added newly.

| | |
|---|---|
| 167 | presence of outliers. |
| 168 | |
| 169 | 2) ITEM INDEPENDENCY ANALYSES |
| 170 | Considering the dynamic approach regarding real-time systems, excluding the *IOI* in the users' |
| 171 | statistical calculations depends on the item's independence condition. Therefore, each particular |
| 172 | item in the datasets was analyzed based on the independence. The leave-item-out approach |
| 173 | emerges as a useful method because the items are independent of each other. Consequently, an |
| 174 | item-based one-way analysis of variance (ANOVA) was performed. Each column (i.e., each |
| 175 | item) is subjected to testing in the *user × item* matrix, validating their independencies. The |
| 176 | ANOVA provides information about inter- and intra-group variations. By calculating the sum of |
| 177 | squares (SS), degrees of freedom (df), and mean squared errors (MS), the F-test (the ratio of |
| 178 | inter- and intra-group variability) is applied. Considering *Table 3 and 4*, the analysis of the |
| 179 | ML100K and ML1M releases is presented. The validity of item independence is proven by both |
| 180 | the F>>1 and probability (P) values which are obtained from the F-distribution. The lower the P- |
| 181 | values are, the higher the chances of a strong evidence against the *null hypothesis*. The P-values |
| 182 | <<0.05 (significance level) are obtained, indicating that the *null hypothesis* is rejected. |
| 183 | |
| 184 | **B. SIMILARITY AND PREDICTION EQUATIONS** |
| 185 | The four touchstone similarity equations and prediction formula are considered in this section. |

*Experimental design #2*

*It is unclear how the median is considered to reduce the outliers; standard techniques to identify (like checking residuals and standardized residuals) should be compared. The median values are indeed better than the mean under certain scenarios, however, it is best to recall, that there are no general unbiased estimators for determining the population median.*

*Note that the Pearson correlation is based on the mean and so the estimator is actually an unbiased estimate of the population statistics.*

## Author Response

First of all, many thanks for your valuable comments. We would like to state that we totally agree with the points you mentioned and have completed our updates on this.

In fact, the use of the median is an approach that we have included in our study to compare different correlation equations. In this context, utilizing MRC in addition to the well-known Pearson similarity equation (with the median rather than the mean) brings a different perspective in terms of performance monitoring and comparison. As you pointed out, the superiority of the median is especially true when it comes to the outlier. Based on this, we performed the residual analysis you suggested on the datasets used. Since this manuscript mainly focuses on static and dynamic approaches, the effect of dynamicity is also measured using the statistical parameter observation thanks to the visualization of residuals. Thereby, the outliers for the MovieLens dataset are interpreted over the residual plots as given on the next page, *Response Figure 1*. They are also newly placed in the revised manuscript as Fig. 1 and Fig. 2.

Our analyses are based on the rating vector of each user. The x-axis of the graphs shows the user IDs, and all users in the dataset have been analyzed. Unique rating values are presented on the y-axis. For any user, it is observed how the statistical values of all the ratings change when an item is assumed not rated. When item-of-interest is not included, statistical calculation of the vector is analyzed within the residual approach according to the situation in which it is included. Analyses for each rating unit (1,2,3,4,5) are presented separately as vertical points projecting on the y-axis. Accordingly, the static value is obtained from all vector elements and is shown in red dots on the plots. Small blue dots show deviations regarding unit vote values. The blue dot count on the vertical axis for a user is equal to the count of unique values in her/his rating vector. Users with five blue dots have at least one unit vote value in their vote history. Visually, the superiority of the median from the residual analyses in both ML100K and ML1M releases is clearly seen in terms of suppressing outliers. The unit rating-related deviation points (blue dots) converge to the corresponding red dot and aggregate, thus depicting the suppression of outliers.

# Response Figure 1: User-based mean & median residuals on static vs. dynamic conditions of ML100K and ML1M

| ML100K | ML1M |
|---|---|

[Figure]

All in all, the related explanations were added to revised manuscript, *Materials & Methods Section*, shown as follows.

144   encapsulate extensive experiments of ML1M to maintain full-star rating scaling parallelism with
145   the ML100K. Therefore, we comparatively present the results related to the original ML100K
146   and ML1M.
147   In this section, preliminary dataset analyses are presented. To validate the methods used in the
148   following sections, residual checks on user-based statistical arguments and item-based
149   independence analyses are performed as follows.
150
151   1) CHECKING RESIDUALS ON USER-BASED STATISTICAL ARGUMENTS
152   The dynamicity effect of user-based statistical arguments (such as the mean and median) is
153   discussed in this subsection. As static and dynamic approaches are the main focus, a
154   visualization of their residual analysis on the utilization of the arguments is presented. Therefore,
155   the rating history of each user was examined based on the statistical observations. Considering
156   any user, it is observed that the statistical values of all the ratings change when an item is
157   assumed as unrated. The *IOI* values that are individually excluded from the user vector are
158   dynamically processed. The effect of each discarded rating was recorded as a residual over the
159   dynamic mean or median. Thereafter, the static observation and dynamic approach are evaluated
160   using the residual approach.
161   *Figure 1* and *2* show the static and dynamic analyses based on the (A) mean and (B) median
162   usage based on the ML100K and ML1M releases, respectively. The x- and y-axis show the user
163   ID and unique rating values, respectively. Whereas each red dot statically gives the mean or
164   median values of all user ratings, the blue dots show the deviation of the unit ratings from the
165   static value. It is observed in the median analysis that the blue dots aggregate, and the outliers in
166   the datasets are suppressed, indicating the superiority of the median over the mean in the
167   presence of outliers.
168
169   2) ITEM INDEPENDENCY ANALYSES
170   Considering the dynamic approach regarding real-time systems, excluding the *IOI* in the users'
171   statistical calculations depends on the item's independence condition. Therefore, each particular
172   item in the datasets was analyzed based on the independence. The leave-item-out approach

*Validity of the findings*

*It is unclear why the smooth coloring of the JAC with SW is of merit. In general, the distribution of metrics is unclear as a statistic of interest, as it is sensitive to the order of the table and the dataset. If the question is of tracking the metrics themselves, then it would be better to describe the data in terms of a density plot.*

## Author Response

Thank you very much for this warning. First of all, our main aim was not to compare the metrics but to compare the different similarity equations over the outstanding neighborhoods, thereby comparing the approaches such as *dynamicity* and *SW*. The relative comparison of the metrics is already given as line plots in the previous sections (Fig. 3-7 in the revised manuscript).

The performance of the methods within themselves is evaluated with the presented heat-map tables (Tables 10 and 11 in the revised manuscript). For this reason, it is not the comparison of columns in the horizontal direction (between metrics) but the comparison of similarity methods in the vertical direction, taking into account the neighborhoods. In this context, each metric was analyzed independently along the relevant column and colored over full precision values. In order to clarify your issue, the minimum and maximum values for each metric where the coloring is performed have been added to the bottom lines of Tables 10 and 11 in the revised manuscript. Each coloring should be evaluated within its own column. That is, the same color may correspond to different values in other columns, but only the single column should be considered to interpret the colorings for any metric.

| $JAC^{SW}_{nlt}$ | | | | | | | | | | | | | | | | | | | | | | | |
|---|---|---|---|---|---|---|---|---|---|---|---|---|---|---|---|---|---|---|---|---|---|---|---|
| 36 | 379 | 364 | 372 | 748 | 725 | 1.039 | 1.020 | 303 | 303 | 550 | 413 | 697 | 682 | 749 | 422 | 598 | 719 | 660 | 780 | 584 | 416 | 220 | 281 | 340 |
| 44 | 379 | 364 | 372 | 748 | 725 | 1.038 | 1.019 | 303 | 303 | 550 | 413 | 697 | 682 | 748 | 422 | 597 | 720 | 660 | 779 | 586 | 414 | 221 | 280 | 340 |
| 45 | 379 | 364 | 372 | 748 | 725 | 1.038 | 1.019 | 303 | 303 | 550 | 413 | 697 | 682 | 748 | 422 | 597 | 720 | 659 | 778 | 586 | 414 | 222 | 280 | 341 |
| 59 | 377 | 363 | 370 | 747 | 726 | 1.038 | 1.019 | 304 | 304 | 551 | 412 | 696 | 682 | 747 | 422 | 596 | 720 | 657 | 775 | 588 | 412 | 225 | 280 | 343 |
| 100 | 373 | 361 | 367 | 744 | 729 | 1.042 | 1.021 | 306 | 306 | 553 | 409 | 694 | 681 | 745 | 422 | 593 | 720 | 653 | 770 | 591 | 409 | 230 | 280 | 347 |
| min | 353 | 339 | 352 | 728 | 719 | 1.024 | 1.012 | 296 | 296 | 544 | 397 | 684 | 670 | 728 | 415 | 572 | 703 | 628 | 732 | 540 | 374 | 201 | 270 | 332 |
| max | 393 | 379 | 386 | 753 | 761 | 1.147 | 1.071 | 316 | 316 | 562 | 420 | 704 | 690 | 753 | 431 | 604 | 730 | 668 | 799 | 626 | 460 | 268 | 297 | 372 |

The fractional values in the table are displayed based on three significant digits. The heat-map coloring is achieved according to full precision.

All in all, we now updated the related paragraph in *Results and Discussion Section* as follows.

579    the tables, highlighting the main motivation of our study: *the decision of adequate weight-metric*

580    *combinations*. Each metric is processed through column-wise coloring to make the comparison

581    easier. Therefore, each coloring is evaluated within its own column. This indicates that the same

582    color may correspond to different values in other columns; nevertheless, only a single column

583    should be considered to interpret the coloring for any metric. The comparison of the similarity

584    methods in the vertical direction is targeted, considering the neighborhoods. At the end of each

585    heat-map table, the minimum and maximum values referenced in the coloring of the relevant

586    column are shown. The tables demonstrate the comparison by addressing the different

587    correlation equations over the outstanding neighborhoods; thereby, comparing approaches such

588    as dynamicity and *SW*, considering each independent metric. The cells shaded in green indicate

589    the effectiveness of the appropriate combination. We present the results using both the *SW*-

590    induced dynamic equations and plain dynamicity; hence, the effect of weight boosting is

591    monitored.

592    All the other test outcomes are found in our code repository[3]. We prepared a fully detailed

In the light of this coloring information, we now explain what the homogeneity of the tables indicates. Since the neighborhood calculation makes the tests dependent on a parameter in recommendation systems, it can be said that the performance of the correlation is better if it is less dependent on neighboring users. That is, a homogeneous method performance, i.e., smooth coloring, is indicated for different best neighbor counts (BNC), so the recommendation algorithm exhibits a less dependent performance. For this reason, the stability of *JAC* with *SW* had been underlined in the first version manuscript. A more specific explanation was now added in the revised manuscript as follows (*Results and Discussion Section*).

600    group is the rest of the metrics, which are frequently used in the literature, including

601    interdisciplinary applications. This method of representation determines the consistency of each

602    similarity equation, considering the groupings. Furthermore, because the tables are multi-

603    dimensional, they include metrics, correlation methods, and multiple parameters such as *BNC*,

604    dynamicity, and *SW*. Because the neighborhood calculation makes the tests dependent on a

605    parameter in recommendation systems, the performance of the correlation is better if it is less

606    dependent on the neighboring users. Therefore, column-wise homogeneity indicates less

607    dependence on the *BNC* range. Remarkably, the homogeneous column-wise scoring highlights

608    the *BNC*-free performance of any similarity equation. For instance, the *JAC* equation with *SW*

609    generally maintains its stability in each metric group, considering smooth coloring. Remarkably,

610    homogeneous scoring highlights the overall performance of any similarity equation.

611    In a general view, if an RS design targets only the recommendations of preferable items, the *COS*

612    may be a suitable similarity measure. Metrics that do not address *TN* values such as *F1-measure*,

***Other Comments from Reviewer - 2***

*The data and code is provided, and this is commendable.*

*The research question posed is appropriate and the code is sufficient for the analysis of the same. However, the manuscript at this stage is not ready for publication and needs to be reworked for clarity. Several key points raised in section 2 in particular need to be addressed adequately. The paper contains implementation details and shows significant effort in terms of covering the existing metrics. I am certain it will be a suitable addition to the literature after revisions.*

## *Author Response*

Finally, we would like to thank you for this motivating comment and all your other feedback. To be honest, we felt that our article was getting more impressive, clearer & more precise while processing all of your feedback. Thank you for taking the time for your valuable suggestions. We hope that our manuscript is now ready for publication in its current state.

---

## Round 0.3 · accepted · Accept

Both reviewers are happy with the revision. As a result, the paper is ready to be accepted.

·

Basic reporting

No comment.

Experimental design

No comment.

Validity of the findings

No comment.

Additional comments

No comment.

Reviewer 3 ·

Basic reporting

The structure and readability of the manuscript have been improved a lot.

Experimental design

The procedure of the experiment and the settings of the parameters are described in sufficient detail.

Validity of the findings

no comment

Additional comments

The revised manuscript has been greatly improved and is suitable for publication.